# Self-Diagnosing GAN: Diagnosing Underrepresented Samples in Generative Adversarial Networks

**Jinhee Lee**[*]
School of Electrical Engineering
KAIST
jin.lee@kaist.ac.kr

**Haeri Kim**[*†]
Samsung Research
Samsung Electronics
haeri.kim@samsung.com

**Youngkyu Hong**[*†]
NAVER AI Lab
NAVER
youngkyu.hong@navercorp.com

**Hye Won Chung**[‡]
School of Electrical Engineering
KAIST
hwchung@kaist.ac.kr

## Abstract

Despite remarkable performance in producing realistic samples, Generative Adversarial Networks (GANs) often produce low-quality samples near low-density regions of the data manifold, e.g., samples of minor groups. Many techniques have been developed to improve the quality of generated samples, either by post-processing generated samples or by pre-processing the empirical data distribution, but at the cost of reduced diversity. To promote diversity in sample generation without degrading the overall quality, we propose a simple yet effective method to diagnose and emphasize underrepresented samples during training of a GAN. The main idea is to use the statistics of the discrepancy between the data distribution and the model distribution at each data instance. Based on the observation that the underrepresented samples have a high average discrepancy or high variability in discrepancy, we propose a method to emphasize those samples during training of a GAN. Our experimental results demonstrate that the proposed method improves GAN performance on various datasets, and it is especially effective in improving the quality and diversity of sample generation for minor groups.

## 1 Introduction

Generative Adversarial Networks (GANs) have achieved remarkable performance in producing realistic samples for complex generation tasks, including image/video synthesis [5, 22], style transfer [45, 14], and data augmentation [29]. However, GANs often fail to cover sparse regions of data manifold [16, 9], leading to the underrepresentation of minor groups in the dataset [43]. In particular, GANs generate samples of minor groups with low fidelity or even fail to generate such samples, exhibiting the mode collapse [43].

Many of previous techniques have focused on improving the overall sample quality of GANs, either by pre-processing the training dataset or by post-processing generated samples. The pre-processing aims to remove instances that cannot be well-represented by GANs even before the training starts and gains fidelity on the focused samples [9]. A similar idea has been used to truncate the latent space by resampling or moving samples that fall outside of some acceptable range during training [16, 6].

---

[*]Equal contribution.  [†]This work was done as a student at KAIST.  [‡]Corresponding author.

35th Conference on Neural Information Processing Systems (NeurIPS 2021).

Post-processing, on the other hand, is a technique that can be applied after the training to remove low-quality generated samples by rejection sampling [3, 37]. All these approaches are effective in increasing the overall fidelity of samples from GANs, but reducing the diversity as a trade-off, and may exacerbate biases against the minor groups in sample generation.

In this work, we aim to improve diversity in sample generation without degrading the overall quality, with a special focus on coverage and quality improvement for minor groups. Toward this, we design methods to detect and emphasize underrepresented samples in training of GANs. Due to the lack of explicit labels available, detecting minor-subgroup samples is especially challenging for unsupervised learning. Therefore, we first develop two new metrics, which can be easily calculated from a discriminator output of GANs, to detect underrepresented samples. The main idea is to measure the statistics (mean and variance) of the estimated discrepancy between the data distribution and model distribution at each data instance over multiple epochs of the training. The mean discrepancy indicates how close the data distribution is to the model distribution at each data over the training, while the variance in discrepancy measures how such discrepancy fluctuates across the training. We provide theoretical and empirical evidence that the mean discrepancy can effectively detect underrepresented samples, especially near collapsed modes, while the variance in discrepancy can detect minor data instances, which GANs suffer from modeling.

Based on these observations, we propose a novel method to emphasize underrepresented samples during the training of GANs by score-based weighted sampling, where the score is defined as a weighted sum of the two metrics we devised. We validate our method with thorough experiments over controlled and real datasets and demonstrate the efficacy of the proposed sampling method in improving not only the overall quality (both fidelity and diversity combined) of sample generation but also the coverage and quality for semantic features of minor subgroups. Our contributions can be summarized as follows.

- We propose two new metrics, which can be simply computed from the discriminator, to diagnose GAN training and to detect underrepresented samples. By theoretical analysis and controlled experiments, we demonstrate that the proposed metrics are effective in detecting underrepresented minor samples.
- We propose an algorithm that can effectively emphasize underrepresented data by score-based weighted sampling during the training of GANs. Our experiments on controlled and real datasets show that our method improves diverse performance metrics on several GAN variants and enhances the coverage and quality of minor group generation.

Our code is publicly available at https://github.com/grayhong/self-diagnosing-gan.

## 2   Related Work

**Promoting data coverage in GANs**   Due to the unstable nature of the min-max game between a generator and a discriminator, GANs often suffer from mode collapse and produce samples with poor diversity. Several approaches have been proposed to promote better data coverage by modifying architectures [20, 23], loss functions [2, 1] or adding regularizations [8, 4, 35]. While effective in promoting overall data coverage, these approaches do not provide special care on minor modes and often fail to recover them when the minority ratio for certain feature is extremely low. We provide a method to promote data coverage for minor features even when the minority ratio is significantly low.

There exists another line of works to improve data coverage by designing hybrid generative models [32, 27, 43], which combines the idea of reconstructive models (e.g. variational autoencoder) to GANs, to take advantages of the reconstructive models in recovering diverse modes. This hybrid method, however, requires relatively high computational overhead to guarantee data coverage for all (or partial) real modes by optimizing reconstruction error in feature domain. Our method directly detects and emphasizes underrepresented samples so that the computational overhead is much lower.

**Improving GAN performance by diagnosing samples**   There have been promising attempts to improve GAN training by using the discriminator outputs to estimate the discrepancy between the data distribution and implicit model distribution. DRS [3] proposes the density ratio estimate based on the discriminator output to apply rejection sampling to filter generated samples. GOLD [26] uses the similar estimate to re-weight fake samples to emphasize underrepresented fake samples. In [10]

and [12], on the other hand, an external classifier is used to improve the density ratio estimates. There also exist some approaches to use discriminator outputs to select or weight "useful" fake samples during training. Top-k training [30] updates the generator by using only top-$k$ fake samples with the largest discriminator outputs. In [31] and [40], discriminator-based importance re-weighting schemes for fake samples are developed, and in [39], latent samples are optimized to improve the fidelity.

Our method uses the discrepancy estimate proposed in [3], but its empirical mean and variance over multiple epochs, to extract more reliable and useful information to detect underrepresented minor group samples. We provide theoretical evidence of why not only the mean but also the variance of discrepancy estimate is effective in detecting underrepresented samples. Our method detects and emphasizes underrepresented real samples, not the fake samples. This difference is significant in promoting the data coverage of minor groups, since when fake samples already fail to cover minor modes, emphasizing a subset of fake samples cannot improve the data coverage for missed modes.

## 3 Two Metrics to Detect Underrepresented Samples During GAN Training

### 3.1 Measuring the discrepancy of GANs

GAN training aims to train a generator with an implicit model distribution $p_g(x)$ that closely matches the data distribution $p_{\mathsf{data}}(x)$. The discrepancy between $p_{\mathsf{data}}(x)$ and $p_g(x)$ can be measured by the log density ratio $\log(p_{\mathsf{data}}(x)/p_g(x))$, but it cannot be directly calculated in GANs, since $p_{\mathsf{data}}(x)$ is unknown and $p_g(x)$ is implicit. Instead, the analysis in the original GAN paper [11] can be used to define an estimate on the density ratio by using the discriminator output as explained in [3].

The original GAN solves the min-max optimization $\min_G \max_D V(D, G)$ for the loss $V(D, G) = \mathbb{E}_{x \sim p_{\mathsf{data}}}[\log D(x)] + \mathbb{E}_{z \sim p_z}[\log(1 - D(G(z)))]$. For any fixed generator $G$, the optimal discriminator yields $D^*(x) = \frac{p_{\mathsf{data}}(x)}{p_{\mathsf{data}}(x) + p_g(x)}$ and this allows us to define the Log-Density-Ratio estimate (LDR) by

$$\mathsf{LDR}(x) := \log(D(x)/(1 - D(x))). \tag{1}$$

When $D(x) = D^*(x)$, the $\mathsf{LDR}(x)$ is equal to the log density ratio $\log(p_{\mathsf{data}}(x)/p_g(x))$. When $\mathsf{LDR}(x) > 0$, the data point $x$ is underrepresented in the model, i.e., $p_{\mathsf{data}}(x) > p_g(x)$, while when $\mathsf{LDR}(x) < 0$, the data is overrepresented, i.e., $p_{\mathsf{data}}(x) < p_g(x)$. Thus, we can leverage the value of $\mathsf{LDR}(x)$ of each instance $x$ to give feedback to improve the generator if the estimation is valid.

Some prior works have used the LDR estimate to improve GAN training. As an example, GOLD [26] uses $\mathsf{LDR}(x)$ to evaluate the quality of the fake samples and re-weights the underrepresented fake samples when training the generator for conditional GANs. However, we later show that re-weighting fake samples is less effective than re-weighting real samples in improving diversity in sample generation. We also empirically show that $\mathsf{LDR}(x)$ is an unstable metric to use. More detailed arguments are available in the Appendix §A.

As a remedy, we propose to use statistics of $\mathsf{LDR}(x)$, which are much more stable and informative metrics, to detect underrepresented data regions during the training. The main intuition is to use training dynamics–the behavior of a model as training progresses–to diagnose the learning behavior of each sample. In supervised learning, training dynamics have been widely studied to detect "hard-to-learn" samples [7, 33, 38]. However, in learning generative models, the metrics to diagnose training dynamics are not clear since there is no explicit reference to measure the accuracy of the model. Here we define metrics that estimate the mean and variance of the discrepancy of GANs, LDRM (LDR Mean) and LDRV (LDR Variance), at each sample $x$ across the training steps $T = \{t_s, ..., t_e\}$:

$$\mathsf{LDRM}(x; T) = \frac{1}{|T|} \sum_{k \in T} \mathsf{LDR}(x)_k, \quad \mathsf{LDRV}(x; T) = \frac{1}{|T| - 1} \sum_{k \in T} \left[\mathsf{LDR}(x)_k - \mathsf{LDRM}(x; T)\right]^2,$$
$$\tag{2}$$

where $\mathsf{LDR}(x)_k$ is the recorded LDR estimate (1) in the $k$-th training step. $\mathsf{LDRM}(x)$ measures how close $p_{\mathsf{data}}(x)$ is to $p_g(x)$ over the training at sample point $x$, while $\mathsf{LDRV}(x)$ measures how such discrepancy fluctuates across training.

Intuitively, samples that have been well-learned and generalized will have consistently small $\mathsf{LDR}(x)$ since $D(x) \approx 1/2$ (i.e., $p_{\mathsf{data}}(x) \approx p_g(x)$), thus will exhibit low LDRM and LDRV, while underrepresented "hard-to-learn" samples will show high LDRM or LDRV values. In the rest of this section, we thoroughly study the characteristics of data instances with high LDRM or high LDRV.

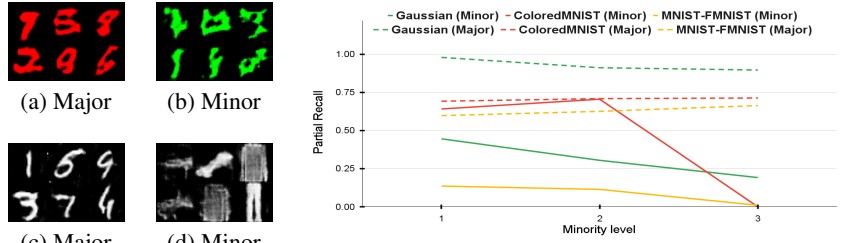

(a) Major     (b) Minor

(c) Major     (d) Minor

(e) Partial Recall of major/minor groups vs. minority level

Figure 1: Analysis on generated samples of GAN trained with (1) Single-mode Gaussian, (2) A mixture of MNIST (major) and FMNIST (minor), and (3) Colored MNIST with Red (major) and Green (minor) samples. (a) $\sim$ (d) show the examples of generated samples with major/minor features, and (e) shows the Partial Recall of major (dotted)/minor (solid) samples in each dataset on various minority levels. Both the sample quality and partial recall rate are higher for major groups.

## 3.2 LDRV is effective in detecting samples from minor groups

**GANs have poor modeling for minor samples**    GANs are known to struggle with modeling minor samples [16]. To scrutinize this phenomenon, we use following toy datasets each of which includes major and minor group: (1) Single-mode Gaussian with distance from the origin as a factor dividing two groups, (2) A mixture of MNIST (major) and FMNIST (minor), and (3) Colored MNIST with Red (major) and Green (minor) digits. We vary the size of the minor group and define a *minority level* to represent the scarcity of the minor group, i.e., a higher level indicates the scarcer minor group. Details of each dataset are available in the Appendix §F. Figure 1 shows the poor quality of generated samples with minor features, relative to major features. To quantify the level of underrepresentedness, we examine the coverage of modes for major vs. minor groups with the Partial Recall [18], which is the portion of the subset of real samples that reside in the manifold of the fake samples. As shown in Figure 1e, major and minor groups have large recall gap and the gap gets worsen as the minority level gets severe. This observation indicates that the minor group suffers not only the poor quality problem but also the low coverage problem, and it gives a strong motivation to detect the minor samples and emphasize them.

**LDRV and minor samples**    We next provide heuristic arguments that LDRV can be used to detect samples with minor features, i.e., features of minor groups. In particular, we show that minor samples tend to have higher LDRV values. First, we view the discriminator as the logistic regression model: for each input $x_i$ the discriminator takes the inner product between the feature vector $\phi_i = F(x_i)$ and the weight vector $\theta$ of the last layer to produce the reality score (the probability that the sample $x_i$ is real ($y_i = 1$)), i.e.,

$$D(x_i; \theta) = 1/(1 + e^{-\theta^T \phi_i}) = p(y_i = 1 | \phi_i, \theta). \tag{3}$$

From a Bayesian perspective, assuming that prior distribution of $\theta$ is $p(\theta) = \mathcal{N}(\theta | 0, s_0 I)$, the posterior distribution over $\theta$ is given by $p(\theta | (\phi_i, y_i)_{i=1}^n) \propto p(\theta) p(y_1^n | \phi_1^n, \theta)$. To obtain a Gaussian approximation to the posterior distribution, we first find the maximum a posteriori estimate $\theta_{\mathsf{MAP}}$ that maximizes $\log p(\theta | (\phi_i, y_i)_{i=1}^n)$, which defines the mean of Gaussian. The covariance is then given by the inverse of the matrix of second derivatives of the negative log likelihood, which takes the form

$$S_n = \left( \sum_{i=1}^n D(x_i; \theta)(1 - D(x_i; \theta)) \phi_i \phi_i^T + \frac{1}{s_0} I \right)^{-1}. \tag{4}$$

Lastly, approximating $D(x_i; \theta)$ in (3) by the Taylor expansion at $\theta = \theta_{\mathsf{MAP}}$, LDRV can be expressed as

$$\mathsf{LDRV}(x_i) \approx \mathsf{var}\left( \log(D(x_i; \theta)/(1 - D(x_i; \theta))) \right) \approx \phi_i^T S_n \phi_i. \tag{5}$$

Details of the analysis is available in the Appendix §B.

This analysis shows an important aspect regarding LDRV and minor features. First, (5) shows that as the feature vector $\phi_i$ becomes more correlated with the principal components of $S_n$ (eigenvectors with largest eigenvalues), its LDRV gets larger. Since each eigenvalue of $S_n$ is the reciprocal of that

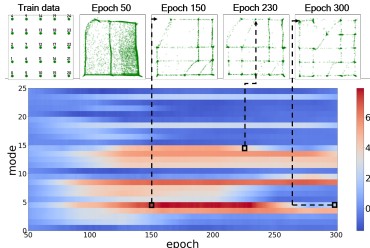
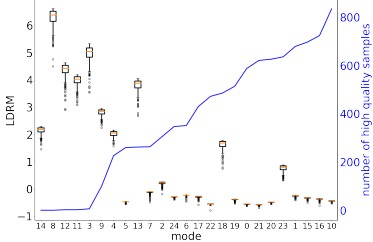

| (a) LDRM and generated samples | (b) LDRM distribution and the number of high-quality samples over modes |
|---|---|

Figure 2: (a) Training dynamics of 25 Gaussians. The index of each mode is equal to $5x + y$ where the coordinates $(x, y) \in \{0, 1, 2, 3, 4\}^2$. LDRM of training samples is recorded for each mode with window $|T| = 50$. Modes with high LDRM do not appear in the generated samples. (b) The empirical distribution of LDRM (box plot) for each mode from training samples, compared with the number of high-quality generated samples from each mode (blue). LDRM can effectively detect dropped modes.

of $S_n^{-1}$, we consider the characteristics of the eigenvector $v$ of $S_n^{-1}$ with the least eigenvalue, which is the minimizer of

$$\langle y, S_n^{-1}y \rangle = \sum_{i=1}^{n} D(x_i; \theta)(1 - D(x_i; \theta))\langle y, \phi_i \rangle^2 + \text{const.} \qquad (6)$$

Eq. (6) shows if $y$ does not align with (or orthogonal to) majority of feature vectors $\{\phi_i\}$ having $D(1 - D) > 0$, then it tends to have a smaller eigenvalue. Since a minor feature vector $\phi_j$ may have a small component on the eigenspace formed by the majority of $\{\phi_i\}$ having $D(1 - D) > 0$, when we plug in $y = \phi_j$ into (6), the summation becomes small. This shows that the minor feature vector $\phi_j$ is correlated with the least eigenvector $v$ of $S_n^{-1}$ and thus it will have higher LDRV.

In Table 1, we show that minor group indeed has higher LDRV. Thus, both theoretical and empirical evidence shows that we can detect minor samples by investigating LDRV of training samples.

### 3.3 LDRM is effective in detecting missing modes

**Mixture of 25 Gaussians** From the definition of LDRM (2), high LDRM samples $x$ tend to have smaller $p_g(x)$ than $p_{\text{data}}(x)$ over the training, thus are underrepresented. We next investigate the ability of LDRM to detect the regions of data manifold not yet covered by the model distribution $p_g(x)$. We consider a mixture of 25 2D isotropic Gaussian distributions [20, 42, 37, 3]. During training, we record $\text{LDR}(x)$ of the training samples and calculate LDRM values with window size $|T| = 50$. We inspect LDRM values averaged over samples of each mode during the training. As shown in Fig. 2a, we observe that samples from underrepresented modes have higher mean LDRM values. This implies that we can detect the mode recovery by inspecting the mean of LDRM values.

To further examine the mode recovery in generated samples, we assign each generated sample to its closest mode and consider it as a "high-quality" sample if it is within four standard deviations from its assigned mode [42, 3]. We then count the number of high-quality samples of each mode among 10,000 generated samples and analyze the correlation between the high-quality sample counts and the distribution of LDRM. As shown in Fig. 2b, modes with only a few high-quality samples tend to have higher LDRM. This indicates that LDRM of the data instances can be used to detect the regions of data manifold not yet covered by the model, even without looking at the generated samples.

Table 1: Averaged LDRV of major/minor groups on various datasets with majority rate 90%.

| Group | Gaussian ($\sigma$=3.0) | Colored MNIST | MNIST-FMNIST |
|---|---|---|---|
| Major | 0.001 | 0.077 | 0.082 |
| Minor | 0.098 | 0.186 | 0.115 |

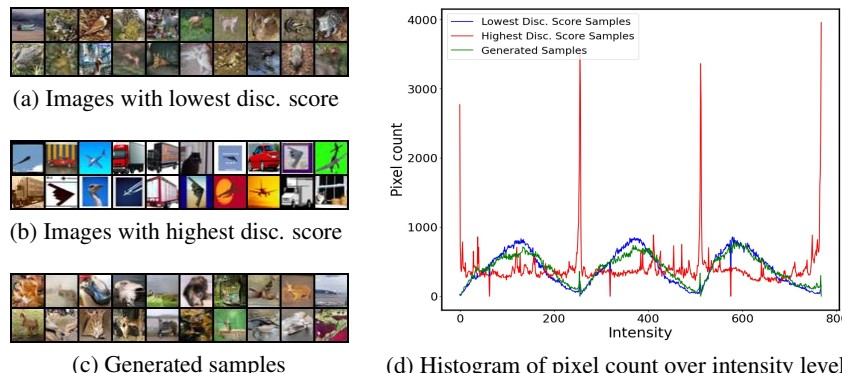

(a) Images with lowest disc. score

(b) Images with highest disc. score

(c) Generated samples

(d) Histogram of pixel count over intensity level

Figure 3: Training images with (a) lowest, (b) highest discrepancy scores, and (c) generated samples. Generated samples resemble training images with lowest score. (d) A smoothed histogram of the intensities for 100 samples per group. The intensity levels of RGB channels are concatenated, resulting in total $768 = 256 \times 3$ levels. Images with the lowest scores (blue) and generated samples (green) have a similar distribution, while images with the highest scores (red) show a high discrepancy.

## 4 Algorithm to Emphasize Underrepresented Samples

### 4.1 Proposed method: Stochastic Gradient Descent (SGD) sampled by discrepancy

We propose a simple modification to the GAN training procedures by using score-based weighted sampling for mini-batch SGD to emphasize underrepresented samples. Let $\mathcal{D} = \{x_i\}$ be the training dataset. The mini-batch of size $B$ for the training dataset is formed by $\mathcal{D}_B = \{x^{(j)} : x^{(j)} = x_i \text{ where } i \sim P_s(i) \text{ for } j = 1, \ldots, B\}$, i.e., each sample $x_i \in \mathcal{D}$ is sampled with certain probability $P_s(i)$. Our objective is to design the sampling frequency $P_s(i)$ that can emphasize underrepresented samples. Based on the observations in Section 3, we first devise the discrepancy score $s(x_i; T)$ that reflects the underrepresentedness of each sample as follows:

$$s(x_i; T) = \mathsf{LDRM}(x_i; T) + k\sqrt{\mathsf{LDRV}(x_i; T)}, \tag{7}$$

where $T$ is the set of steps used to calculate the discrepancy scores and $k$ is the hyperparameter to modulate the contribution of each statistic. The score (7) can be interpreted as an upper limit of the confidence interval of LDR estimate, or weighted sum of LDRM and the square root of LDRV with weight controlled by $k$. To ensure every data is sampled with at least some chance, we clip the minimum value of $s(x_i)$ to be $\epsilon = 0.01$ (min_clip) and clip the maximum value to have max-min ratio of 50, i.e., $\max s(x)/\min s(x) = 50$ (max_clip). For the clipped score $s'(x_i; T) = \mathtt{max\_clip}(\mathtt{min\_clip}(s(x_i; T)))$, our final weighted sampling frequency is $P_s(i) = \frac{s'(x_i; T)}{\sum_{j=1}^{|\mathcal{D}|} s'(x_j; T)}$.

### 4.2 Sample analysis of the discrepancy score

To check whether our discrepancy score indeed captures the underrepresented samples, we analyze the samples with lowest/highest discrepancy scores. We train SNGAN [25] on CIFAR-10 [17] for 40k steps and measure the discrepancy score of each sample. We first present the images with lowest (Fig. 3a)/highest (Fig. 3b) discrepancy scores among training images, and compare them with generated samples (Fig. 3c). High-scoring images have properties that are distinct from the generated samples (e.g., unusual background or shape), while low-scoring images contain features that are also available in generated samples. Comparing the pixel intensity histogram (Fig. 3d) reveals the difference more clearly in sample properties. Images with lowest discrepancy scores exhibit similar intensity distribution with generated samples, while images with highest scores appear to show an extremely different tendency. We also analyze the Partial FID (FID [13] measured with a subset of training samples) of lowest/highest-score groups. The highest-score group has a Partial FID of 94.64 while the lowest-score group has 22.43. The large gap between the two groups states that the generator fails to generate samples similar to high-score group. These results imply that our discrepancy score

successfully identifies underrepresented data that may need emphasis in further training. For more examples of images with scores both for CIFAR-10 and CelebA, see the Appendix §C.

### 4.3 Post-processing by discriminator rejection sampling with auxiliary discriminator

Our weighted sampling gives bias toward underrepresented samples during training. Though effective in improving diversity, this results in modified data distribution $p'_{\mathsf{data}}(x) = f(x)p_{\mathsf{data}}(x)$ where $f(\cdot)$ is the normalized sampling frequency. Thus, the trained model distribution $p_g$ may be different from the original data distribution $p_{\mathsf{data}}$. To solve this, we utilize the Discriminator Rejection Sampling (DRS) [3] to correct the bias after training. The rejection sampling accepts a generated sample with probability $\frac{p_{\mathsf{data}}(x)}{Mp_g(x)}$ for some constant $M > 0$. To conduct rejection sampling, DRS method needs an estimate for $p_{\mathsf{data}}(x)/p_g(x)$ calculated based on the discriminator outputs. Since our discriminator is trained with biased $p'_{\mathsf{data}}(x)$, we add an auxiliary discriminator and train it with uniform sampling (i.e., without applying our sampling technique) during the weighted sampling procedure to obtain the LDR estimate (1) for DRS, and use this measure for the rejection sampling of generated samples.

### 4.4 Self-Diagnosing GAN (Dia-GAN)

The overall algorithm (with details in the Appendix §D) can be summarized as below:
**Phase 1 - Train and Diagnose:** Train GAN and evaluate the discrepancy score for each data instance.
**Phase 2 - Score-Based Weighted Sampling:** Encourage GAN to learn underrepresented regions of data manifold through score-based weighted sampling (Section 4.1).
**Phase 3 - DRS:** After GAN training, correct the model distribution $p_g(x)$ by rejection sampling.

## 5 Experiments

### 5.1 Evaluation metrics and baselines

**Evaluation metrics** To evaluate the effect of our method on learned model distribution, we use various performance metrics including (1) Fréchet Inception Distance (FID) [13], (2) Inception Score (IS) [28], and (3) Precision and Recall (P&R) [18]. In addition to these global evaluation metrics, we consider (4) Reconstruction Error (RE). RE score is calculated by first training a convolutional autoencoder (CAE) with generated samples, and then calculating Euclidean distance between each training data and its reconstruction. RE can assess whether $p_g(x)$ covers $p_{\mathsf{data}}(x)$ since CAE is known to have high RE for out-of-distribution samples [41, 44]. For more details, see the Appendix §E.

**Baselines** We compare the effect of our method with other methods that use the discriminator output for improving GAN training; 1) DRS [3], 2) Gap of log-densities (GOLD) [26], and 3) Top-k training [30]. GOLD[4] uses the LDR estimate on generated samples to re-weight underrepresented samples (having high LDR) during training of GANs. Top-k training uses only top-$k$ fake samples with the largest discriminator outputs, i.e., the samples believed to be the "most realistic", during the training of the generator. As our algorithm uses DRS after the training, we also analyze each method's performance with post-processing by DRS to measure the exact gain from our sampling method.

### 5.2 GAN performance enhancement on real datasets

**Experiments on CIFAR-10 and CelebA** We first assess our method on two widely-studied GAN benchmark datasets, CIFAR-10 [17] and CelebA [21]. We evaluate our method on state-of-the-art GANs; SNGAN [25] and SSGAN [36] with non-saturating variant of the original loss. We train our model for 50k (75k) steps for CIFAR-10 (CelebA), where for our method and GOLD, the phase 1 takes 40k (60k) steps, and the phase 2 takes the remaining. We record LDR every 100 steps and use the last 50 records for calculating the discrepancy score. For the discrepancy score (7), we use $k = 0.3$ (5.0) for CIFAR-10 (CelebA). Detailed configurations and hyperparameter search procedure are available in the Appendix §F.

---

[4] As the original GOLD estimator is designed for conditional GANs [24], we consider the unconditional version by removing the conditional discrepancy term.

Table 2: Comparison of diverse sampling/weighting methods for CIFAR-10/CelebA image generation.

| Dataset | CIFAR-10 | | | | CelebA | | | | | |
|---|---|---|---|---|---|---|---|---|---|---|
| Methods | SNGAN | | SSGAN | | SNGAN | | | SSGAN | | |
| | FID ↓ | IS ↑ | FID ↓ | IS ↑ | FID ↓ | P ↑ | R ↑ | FID ↓ | P ↑ | R ↑ |
| Vanilla | 26.90 | 7.36 | 22.01 | 7.65 | 7.12 | 0.68 | 0.44 | 7.19 | 0.68 | 0.44 |
| DRS [3] | 24.54 | 7.57 | 20.51 | 7.77 | 7.04 | 0.68 | 0.44 | 7.08 | 0.68 | 0.45 |
| GOLD [26] | 28.86 | 7.21 | 21.90 | 7.57 | 7.31 | **0.69** | 0.44 | 7.46 | 0.68 | 0.43 |
| GOLD + DRS | 24.65 | 7.53 | 19.36 | 7.79 | 6.97 | 0.68 | 0.44 | 7.15 | 0.67 | 0.45 |
| Top-k [30] | 24.45 | 7.60 | 20.01 | 7.78 | 7.35 | 0.67 | 0.44 | 7.23 | 0.67 | 0.45 |
| Top-k + DRS | 23.92 | 7.70 | 20.09 | 7.88 | 7.35 | 0.68 | 0.44 | 7.16 | **0.68** | 0.45 |
| **Dia-GAN** | **19.66** | **7.95** | **16.31** | **8.14** | **6.70** | 0.64 | **0.48** | **6.88** | 0.66 | **0.46** |

Table 3: StyleGAN2 on FFHQ 256x256.

| | FID ↓ | P ↑ | R ↑ |
|---|---|---|---|
| StyleGAN2 | 14.07 | **0.72** | 0.27 |
| GOLD | 15.53 | 0.69 | 0.29 |
| **Dia-StyleGAN2** | **11.89** | 0.69 | **0.30** |

Table 4: HingeGAN on CIFAR-10 and CelebA.

| | CIFAR-10 | | CelebA |
|---|---|---|---|
| | FID ↓ | IS ↑ | FID ↓ |
| HingeGAN | 21.99 | 7.67 | 6.66 |
| **Dia-HingeGAN** | **18.74** | **8.02** | **5.98** |

In Table 2, we first compare FID and IS over various methods on the CIFAR-10 dataset. Our proposed Dia-GAN achieves the best FID and IS with a great margin among all baseline methods in every GAN variant. This result demonstrates the wide applicability and effectiveness of our method in improving the overall quality (fidelity and diversity combined) of generated samples. Moreover, the comparison between DRS and our method assures that most of the gain indeed comes from our resampling method. Also, we compare FID and P&R over the methods on CelebA. Our method consistently improves FID over baseline GANs. Precision & Recall analysis shows more detailed reasons for the improvement of FID. Our method consistently improves recall (diversity) but with a slight drop in precision (fidelity). As the increase in diversity is dominant, FID, which measures the combined effect of fidelity and diversity, is consistently improved with our method compared to the baselines. Examples of generated samples from our Dia-GAN are also available in the Appendix §G.

**Experiments on StyleGAN2** We further evaluate the scalability of our method with Style-GAN2 [16] on FFHQ 256x256 [15] dataset. We train the model for 250k steps in total where phase 1 takes 200k steps and the phase 2 takes the remaining steps. We set the hyperparameter $k = 3.0$. Our method improves the FID of StyleGAN2 from 14.07 to 11.89 and recall of StyleGAN2 from 0.27 to 0.30 as shown in Table 3. This indicates that our method successfully scales to large state-of-the-art GANs and high-resolution images.

**Extension to hinge loss** We further conduct experiments to show the applicability of our method to other GAN losses. Here, we focus on a commonly used loss, the hinge loss (HingeGAN) [19, 34]. Our method is not directly applicable to the hinge loss since the output of the optimal discriminator $D_h(x)$ is not $\frac{p_{\text{data}}(x)}{p_{\text{data}}(x) + p_g(x)}$ anymore. Instead, $D_h(x)$ is 1 if $p_{\text{data}}(x) > p_g(x)$ and $-1$ if $p_{\text{data}}(x) < p_g(x)$. One possible workaround is attaching an auxiliary layer to the discriminator and training it with the original GAN loss. However, we instead present empirical evidence showing that $D_h(x)$ itself still contains useful information about the degree of learning for the input $x$. We consider the variant of our method, Dia-HingeGAN, by calculating the mean and variance of $D_h(x)$ and using the same scoring rule of (7). In Table 4, we compare the performance of HingeGAN and Dia-HingeGAN with the same configuration of the previous experiment. Interestingly, our method shows significant improvement in both CIFAR-10 and CelebA. This implies that despite the optimal form of the discriminator is different, the statistics of its output still provide meaningful information about the underrepresented features. We leave the theoretical analysis of this variant method as a future work.

Table 5: Reconstruction Error (RE) score of green (minor) training samples in Colored MNIST and FMNIST (minor) samples in a mixture of MNIST and FMNIST on different majority rate $\rho$.

| Dataset | Colored MNIST | | | MNIST-FMNIST | | |
|---|---|---|---|---|---|---|
| Majority rate $\rho$ | 99% | 95% | 90% | 99% | 95% | 90% |
| Vanilla | 0.838 | 0.236 | 0.218 | 0.290 | 0.227 | 0.215 |
| GOLD [26] | 0.813 | 0.297 | 0.200 | 0.296 | 0.241 | 0.218 |
| Top-k [30] | 0.831 | 0.210 | 0.223 | 0.281 | 0.232 | 0.221 |
| PacGAN [20] | 0.810 | 0.244 | 0.233 | 0.313 | 0.251 | 0.225 |
| Inclusive GAN [43] | 0.812 | 0.274 | 0.216 | 0.283 | 0.230 | 0.220 |
| **Dia-GAN** | **0.224** | **0.204** | **0.197** | **0.264** | **0.219** | **0.206** |

Table 6: CelebA minor attribute analysis. Averaged LDRV and averaged discrepancy score of CelebA samples with (W/) or without (W/O) minor attributes. O stands for the occurrence of minor attributes among the generated samples in percentage (%) and R stands for the Partial Recall.

| | Score | | | | Method | | | |
|---|---|---|---|---|---|---|---|---|
| | LDRV | | Discrepancy | | Vanilla | | **Dia-GAN** | |
| | W/ | W/O | W/ | W/O | O ↑ | R ↑ | O ↑ | R ↑ |
| Bald (2.244%) | **0.271** | 0.184 | **2.938** | 2.221 | 0.678 | 0.353 | **0.836** | **0.393** |
| Double Chin (4.669%) | **0.219** | 0.184 | **2.525** | 2.224 | 0.440 | 0.411 | **0.522** | **0.461** |
| Eyeglasses (6.512%) | **0.254** | 0.181 | **2.783** | 2.200 | 3.300 | 0.400 | **4.053** | **0.449** |
| Gray Hair (4.195%) | **0.211** | 0.185 | **2.450** | 2.228 | 2.273 | 0.402 | **2.369** | **0.436** |
| Mustache (4.155%) | **0.242** | 0.183 | **2.699** | 2.218 | 0.157 | 0.391 | **0.228** | **0.433** |
| Pale Skin (4.295%) | **0.190** | 0.186 | **2.240** | 2.238 | 0.346 | 0.380 | **0.453** | **0.427** |
| Wearing Hat (4.846%) | **0.357** | 0.177 | **3.651** | 2.164 | 2.307 | 0.380 | **3.595** | **0.408** |

## 5.3 Minor feature generation

**Controlled experiments**   As our method emphasizes underrepresented samples in GAN training, we evaluate how much our method helps the generation of minor samples. To control the level of minority, we design a Colored MNIST dataset with red (major) and green (minor) samples, and MNIST-FNIST dataset with MNIST (major) FMNIST (minor) samples, with the majority rates $\rho \in \{90, 95, 99\}\%$. We compare our method with the same set of baseline methods as in Section 5.1. Additionally, we compare our method with PacGAN [20], the approach to handle the mode collapse problem, and with Inclusive GAN [43], which also improves the data coverage over the minor groups by using a hybrid generative model.

Table 5 shows the results of each method in various majority rates. Here, we focus on the reconstruction error (RE) score of minor training samples (green samples for Colored MNIST, and FMINST samples for MNIST-FMNIST dataset). For the training dataset with the majority rate of 99%, our method shows a significant improvement in RE score as only our method succeeds in generating minor samples while others fail. When the majority rate $\rho$ decreases to 95% and 90%, vanilla model starts to generate minor samples but in low quality. For these rates, our method also shows improvement on the quality of generated samples with minor features, resulting in better RE scores. This result implies the efficacy of our method in improving the quality of generated samples with underrepresented features. Examples of samples with minor features for each method are available in the Appendix §I, and detailed configuration of the experiment is available in the Appendix §F.

**CelebA minor attribute analysis**   For the real-world example, we analyze how our method changes the generation of minor attributes of the CelebA [21] dataset, using the meta-information available in the dataset. Specifically, we focus on how much our method improves the occurrence rate of minor attributes, as they usually appear in a much lower rate than its actual ratio. We train a binary classifier for each attribute to have train and test accuracy above 95%. We also evaluate the Partial Recall of the minor attributes, since minor samples suffer low-recall problem as explained in Section 3.2.

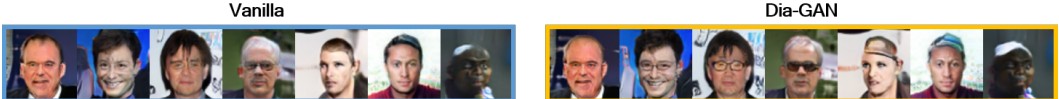

Figure 4: Generated samples of vanilla SNGAN and Dia-SNGAN trained on CelebA.

As shown in Table 6, our method improves both the occurrence (O) and Partial Recall (R) rates of various minor attributes. Moreover, as explained in Section 3.2, minor samples do have higher LDRV (2) and the discrepancy score (7). This indicates that our method indeed captures the underrepresented minor features and successfully promotes the generation of such features during training of GANs. Figure 4 shows examples of generated samples with minor feature appeared by our Dia-GAN. Note that as we use the majority of the training time for Phase 1 (80% of total steps), the generator partially converges after Phase 1 and thus the same latent vector $z$ turns out to give a similar image with details changed after Phase 2 (e.g., wearing sunglasses, or a hat).

The ability of our method in capturing semantic features and improving the generation of minor samples also applies to high-resolution datasets. To demonstrate this, we conduct similar experiments for the high-resolution FFHQ dataset and present the results in the Appendix §H.

## 6 Discussion

We proposed two new metrics, LDRV and LDRM, that can detect underrepresented samples and devised a simple approach to emphasize detected underrepresented samples. Our method successfully improves overall quality of the generated samples in terms of FID and IS, and promotes generation of minor samples. However, we still find the trade-off relationship between the precision and recall of generated samples (Table 2). We leave the investigation of other approaches to use the knowledge of detected underrepresented samples for further improvement of GAN training as a future work.

**Societal impact**   We propose a discrepancy score that can detect underrepresented minor samples in training of GANs. On the good side, this results in enhanced generation of minor samples in GANs. The ability of our score could be further expanded and used for utilizing skewed datasets to train models representing more balanced datasets, by adding a hyperparameter that can tune the level of emphasis for underrepresented samples. On the other hand, an abuser might instead be able to remove such minor subgroup samples and deteriorate the bias in sample generation.

## Acknowledgement

This research was supported by the National Research Foundation of Korea under Grant 2017R1E1A1A01076340 and 2021R1C1C11008539, and by the Ministry of Science and ICT, Korea, under the IITP (Institute for Information and Communications Technology Panning and Evaluation) grant (No.2020-0-00626).

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
