# Appendix for "Self-Diagnosing GAN: Diagnosing Underrepresented Samples in Generative Adversarial Networks"

**Jinhee Lee**[*]
KAIST

**Haeri Kim**[*†]
Samsung Electronics

**Youngkyu Hong**[*†]
NAVER

**Hye Won Chung**[‡]
KAIST

## A  Instability of LDR estimate

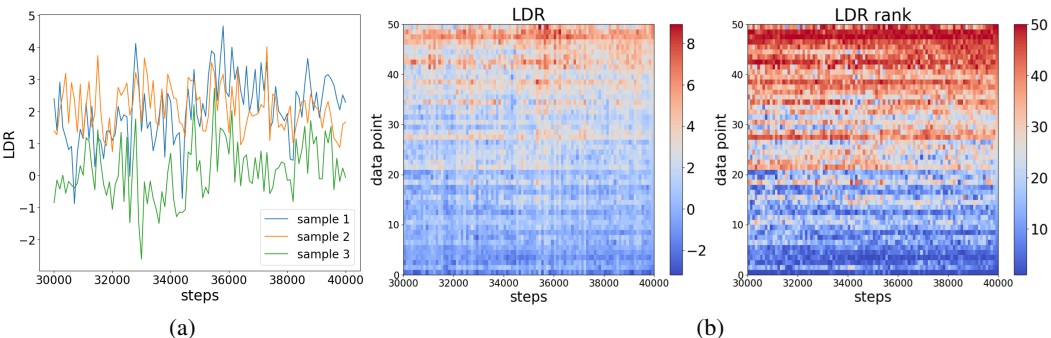

Figure A1: (a) LDR values of three randomly chosen samples, (b) LDR values (left) and rankings (right) of 50 samples during training of CIFAR-10 on SNGAN [15]. The values are recorded every 100 steps from 30000 to 40000 steps (total 100 times). LDR values are unstable during training, so it is hard to diagnose GAN training from the LDR of a particular training step. The level of fluctuations varies much over samples.

The Log-Density-Ratio estimate (LDR) is defined by

$$\mathsf{LDR}(x) := \log \frac{D(x)}{1 - D(x)}. \tag{A.1}$$

When $D(x) = D^*(x)$, the $\mathsf{LDR}(x)$ is equal to the log density ratio $\log(p_{\mathsf{data}}(x)/p_g(x))$. When $\mathsf{LDR}(x) > 0$, the data point $x$ is underrepresented in the model, i.e., $p_{\mathsf{data}}(x) > p_g(x)$, while when $\mathsf{LDR}(x) < 0$, the data is overrepresented, i.e., $p_{\mathsf{data}}(x) < p_g(x)$. Thus, we can leverage the value of $\mathsf{LDR}(x)$ of each instance $x$ to give feedback to improve the generator if the estimation is valid. Some prior works have used this tendency to evaluate the quality of fake samples and designed sample reweighting scheme to guide the generator to focus on underestimated samples [16] or rejection sampling to post-process generated samples to approximately correct errors in the model distribution [1].

The effectiveness of the above schemes highly depends on the accuracy of the LDR estimate. However, we observe that $\mathsf{LDR}(x)$ is unstable during the training even after large steps, as shown in Fig. A1.

---

[*]Equal contribution.

[†]This work was done as a student at KAIST.

[‡]Corresponding author: hwchung@kaist.ac.kr

35th Conference on Neural Information Processing Systems (NeurIPS 2021).

Therefore, to have a better estimate on LDR, we propose to use statistics (mean and variance) of LDR estimates over multiple steps (epochs) of the training. Different from [1, 16], we focus on the discrepancy of GANs at training data instances rather than at generated samples. This leads us to fully explore the underrepresented regions of the data manifold during the training, which can then be emphasized to improve the performance of GANs.

## B  Analysis of variance of LDR estimate

Consider the discriminator trained with a data set $\{(x_i, y_i)_{i=1}^n\}$ to minimize the cross-entropy loss

$$-\sum_{i=1}^n (y_i \log D(x_i) + (1 - y_i) \log(1 - D(x_i))) \tag{B.1}$$

where $y_i = 1$ for a real sample and $y_i = 0$ for a fake sample. Assuming that $\phi_i = F(x_i) \in \mathbb{R}^d$ denotes the feature vector of $x_i$ extracted by the discriminator and that the discriminator is defined by a sigmoid applied to $\theta^T \phi_i$ for some $d$-dimensional parameter $\theta$ just like the logistic regression, the discriminator output can be considered as the probability that the input $x_i$ is a real sample, i.e.,

$$D(x_i; \theta) = \frac{1}{1 + e^{-\theta^T \phi_i}} = p(y_i = 1 | \phi_i, \theta). \tag{B.2}$$

We now turn to a Bayesian treatment of logistic regression and find the Gaussian approximation for the posterior distribution of $\theta$ given the data set, in a similar way as in Section 4.5 of [2]. Assume that

$$p(\theta) = \mathcal{N}(\theta | 0, s_0 I) \tag{B.3}$$

where $s_0$ is a fixed hyperparameter. The posterior distribution over $\theta$ is given by

$$p(\theta | (\phi_i, y_i)_{i=1}^n) \propto p(\theta) p(y_1^n | \phi_1^n, \theta). \tag{B.4}$$

Taking the log of both sides, and substituting for the prior distribution (B.3), and the likelihood function using (B.2), we obtain

$$\log p(\theta | (\phi_i, y_i)_{i=1}^n) = \sum_{i=1}^n (y_i \log D(x_i; \theta) + (1 - y_i) \log(1 - D(x_i; \theta))) - \frac{\|\theta\|^2}{2s_0} + \text{const.} \tag{B.5}$$

for $D(x_i; \theta)$ in (B.2). To obtain a Gaussian approximation to the posterior distribution, we first find $\theta_{\text{MAP}}$ that maximizes $\log p(\theta | (\phi_i, y_i)_{i=1}^n)$, i.e., $\frac{d}{d\theta} \log p(\theta_d | (\phi_i, y_i)_{i=1}^n)\big|_{\theta = \theta_{\text{MAP}}} = 0$, which defines the mean of the Gaussian. The covariance is then given by the inverse of the matrix of second derivatives of the negative log likelihood, which takes the form

$$S_n^{-1} = -\nabla_\theta \nabla_\theta \log p(\theta | (\phi_i, y_i)_{i=1}^n) = \sum_{i=1}^n D(x_i; \theta)(1 - D(x_i; \theta))\phi_i \phi_i^T + \frac{1}{s_0} I. \tag{B.6}$$

The Gaussian approximation of the posterior distribution of $\theta$ thus takes the form of

$$p(\theta | (\phi_i, y_i)_{i=1}^n) \approx \mathcal{N}(\theta | \theta_{\text{MAP}}, S_n). \tag{B.7}$$

We next relate the variance of LDR estimate for each data sample with the covariance matrix $S_n$. First, we can find that

$$\text{var}\left(\log \frac{D(x_i; \theta)}{1 - D(x_i; \theta)}\right) = \text{var}(\log D) + \text{var}(\log(1 - D)) - 2\text{cov}(\log D, \log(1 - D)). \tag{B.8}$$

By approximating $D(x_i; \theta)$ by the Taylor expansion at $\theta = \theta_{\text{MAP}}$, we get

$$\log D(x_i; \theta) \approx \log D(x_i; \theta_{\text{MAP}}) + (1 - D(x_i; \theta_{\text{MAP}}))\phi_i^T (\theta - \theta_{\text{MAP}}),$$
$$\log(1 - D(x_i; \theta)) \approx \log(1 - D(x_i; \theta_{\text{MAP}})) - D(x_i; \theta_{\text{MAP}})\phi_i^T (\theta - \theta_{\text{MAP}}), \tag{B.9}$$

and thus the variances are

$$\text{var}(\log D(x_i; \theta)) \approx (1 - D(x_i; \theta_{\text{MAP}}))^2 \phi_i^T S_n \phi_i,$$
$$\text{var}(\log(1 - D(x_i; \theta))) \approx D(x_i; \theta_{\text{MAP}})^2 \phi_i^T S_n \phi_i, \tag{B.10}$$

for the covariance matrix $S_n$ of (B.7). Using the similar Taylor expansion, we can approximate

$$\mathrm{cov}(\log D, \log(1-D)) = \mathbb{E}[\log D \cdot \log(1-D)] - \mathbb{E}[\log D]\mathbb{E}[\log(1-D)]$$
$$\approx -D(x_i; \theta_{\mathsf{MAP}})(1 - D(x_i; \theta_{\mathsf{MAP}}))\phi_i^T S_n \phi_i. \tag{B.11}$$

By combining the above results,

$$\mathrm{var}\left(\log \frac{D(x_i; \theta)}{1 - D(x_i; \theta)}\right) \approx (D^2 + (1-D)^2 + 2D(1-D))\phi_i^T S_n \phi_i \tag{B.12}$$
$$= \phi_i^T S_n \phi_i.$$

Finally, by plugging in $S_n$, the variance of LDR estimate for each sample $x_i$ with feature vector $\phi_i$ can be written as

$$\mathrm{var}\left(\log \frac{D(x_i; \theta)}{1 - D(x_i; \theta)}\right) \approx \phi_i^T \left(\sum_{i=1}^{n} D(x_i; \theta)(1 - D(x_i; \theta))\phi_i\phi_i^T + \frac{1}{s_0}I\right)^{-1} \phi_i. \tag{B.13}$$

## C  Images with lowest/highest discrepancy score for CIFAR-10 & CelebA

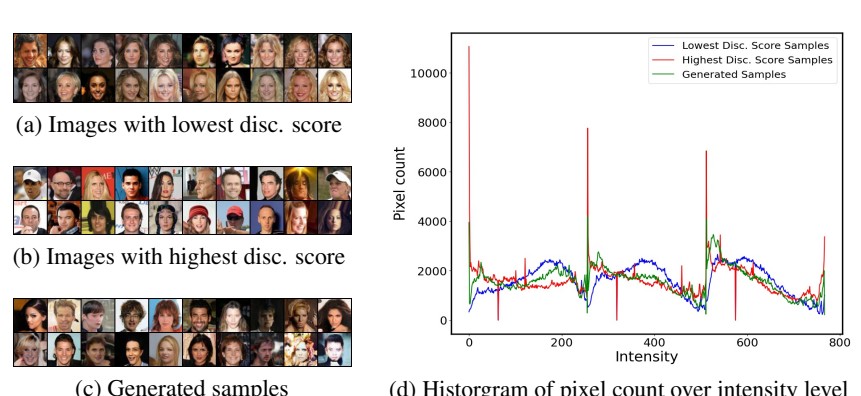

(a) Images with lowest disc. score

(b) Images with highest disc. score

(c) Generated samples

(d) Historgram of pixel count over intensity level

Figure A2: Examples of CelebA training images with (a) lowest and (b) highest discrepancy scores, and (c) generated samples after Phase 1 (without weighted sampling). Generated samples resemble training images with lowest discrepancy score. (d) A smoothed histogram of the intensities for 100 samples per group. The intensity levels of RGB channels are concatenated, resulting in a total of $768 = 256 \times 3$ levels. Images with the lowest scores (blue) and generated samples (green) have a similar distribution, while images with the highest scores (red) show a high discrepancy.

In this section, we show the characteristics of training images having lowest/highest discrepancy scores. As of Fig. 3 (CIFAR-10) in the main paper, we present the images with lowest (Fig. A2a)/highest (Fig. A2b) discrepancy scores among CelebA training images, and compare them with generated samples (Fig. A2c), after Phase 1 of our algorithm (before sample-weighting starts). Comparing the pixel intensity histogram (Fig. A2d) reveals more clearly the difference in sample properties. Images with low discrepancy scores exhibit similar intensity distribution with generated samples, while images with high scores appear to show an extremely different tendency. These results show that our discrepancy score successfully distinguishes underrepresented instances.

We also present the samples with lowest/highest discrepancy scores with various $k$ values (the hyperparameter for discrepancy score (7)) for CIFAR-10 (Figure A3, A4) & CelebA (A5, A6). Samples with high discrepancy scores have properties that are distinct from the samples with low discrepancy scores (e.g. vividness or unusual backgrounds for CIFAR-10 and minor features such as diverse hair colors or accessories including glasses or hats for CelebA). Since generated samples resemble the images with low discrepancy scores, emphasizing high-scoring images can boost the diversity in sample generation.

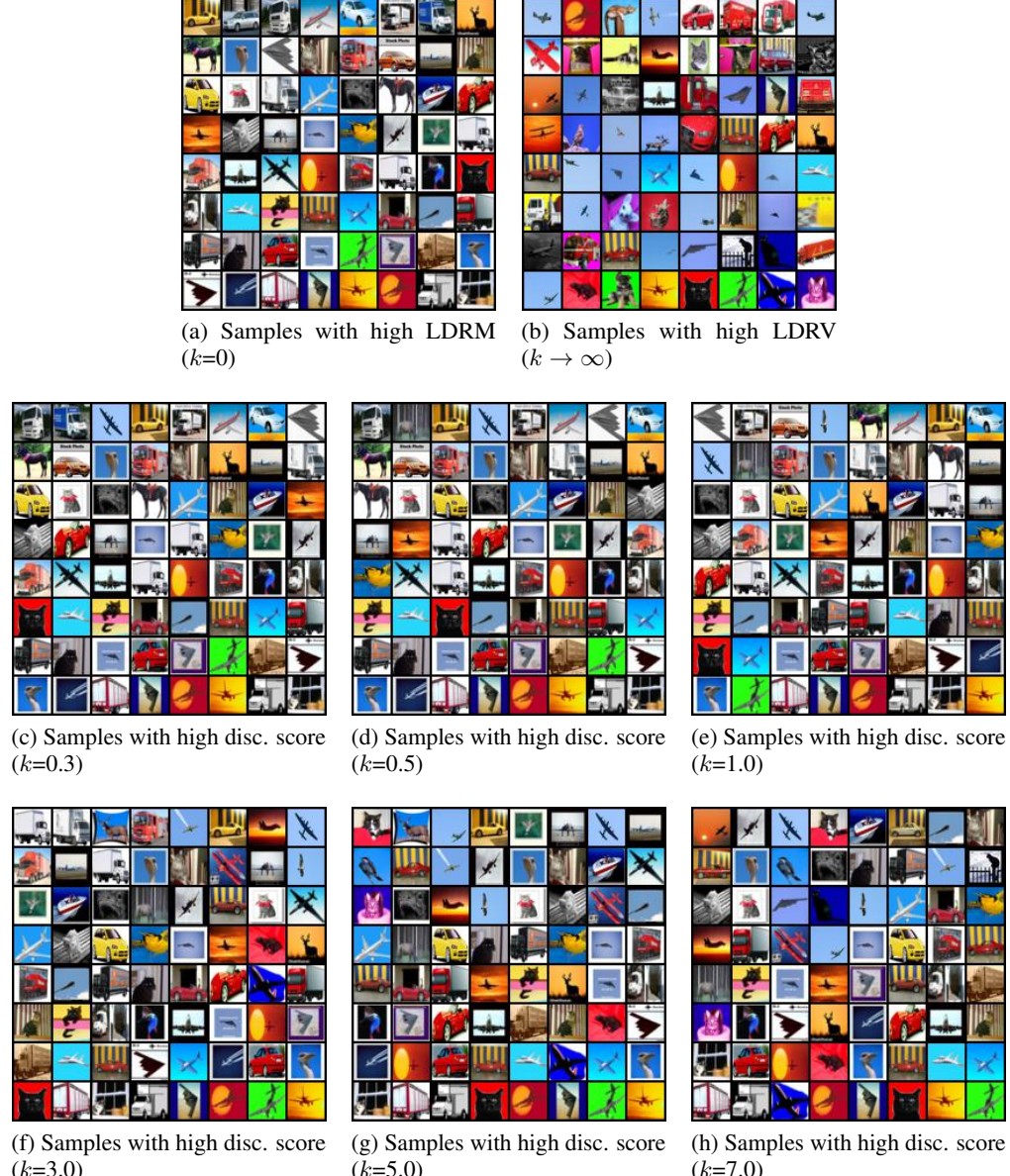

(a) Samples with high LDRM ($k$=0)

(b) Samples with high LDRV ($k \to \infty$)

(c) Samples with high disc. score ($k$=0.3)

(d) Samples with high disc. score ($k$=0.5)

(e) Samples with high disc. score ($k$=1.0)

(f) Samples with high disc. score ($k$=3.0)

(g) Samples with high disc. score ($k$=5.0)

(h) Samples with high disc. score ($k$=7.0)

Figure A3: CIFAR-10 samples with highest discrepancy scores on various $k$

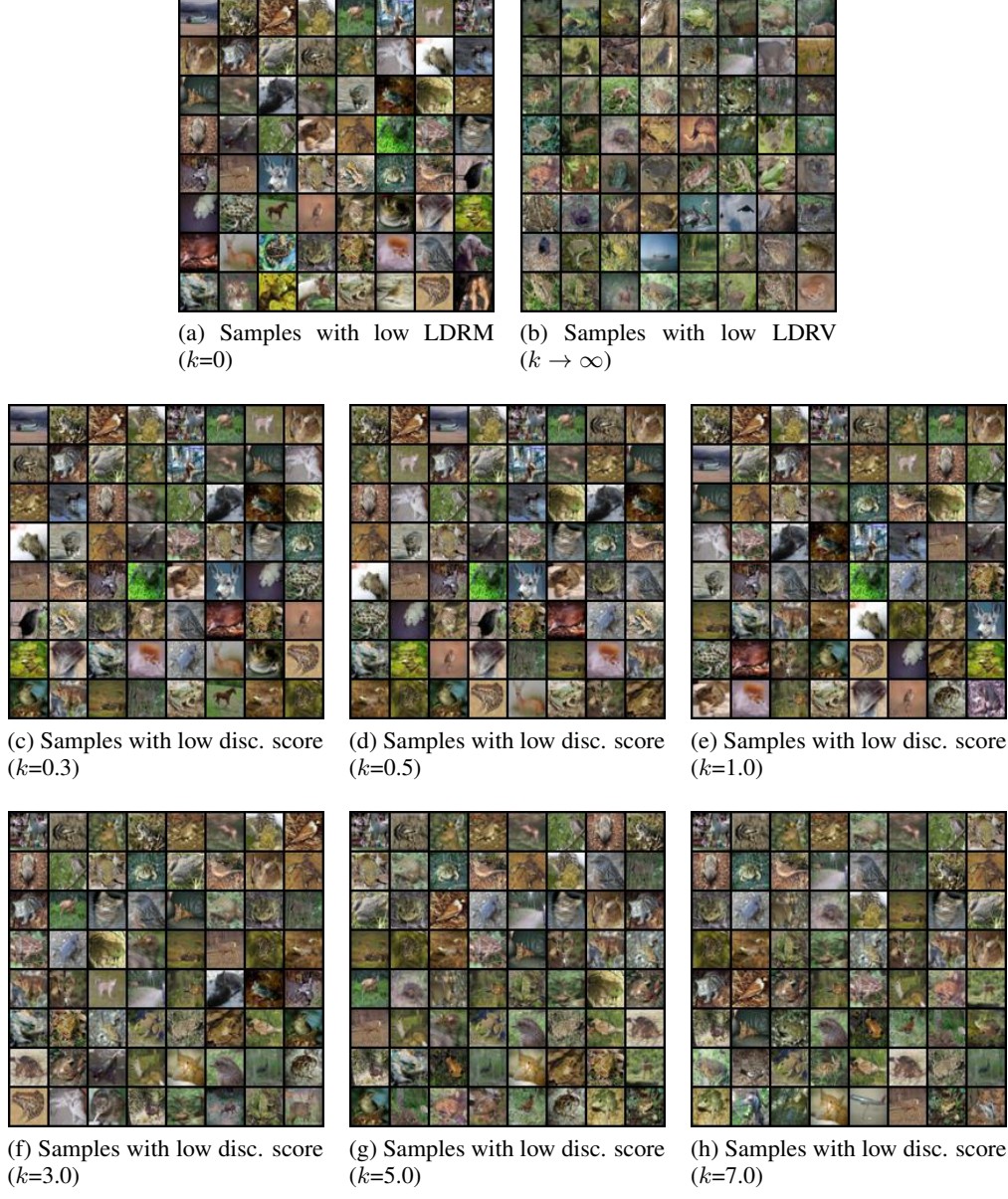

(a) Samples with low LDRM ($k$=0)

(b) Samples with low LDRV ($k \rightarrow \infty$)

(c) Samples with low disc. score ($k$=0.3)

(d) Samples with low disc. score ($k$=0.5)

(e) Samples with low disc. score ($k$=1.0)

(f) Samples with low disc. score ($k$=3.0)

(g) Samples with low disc. score ($k$=5.0)

(h) Samples with low disc. score ($k$=7.0)

Figure A4: CIFAR-10 samples with lowest discrepancy scores on various $k$

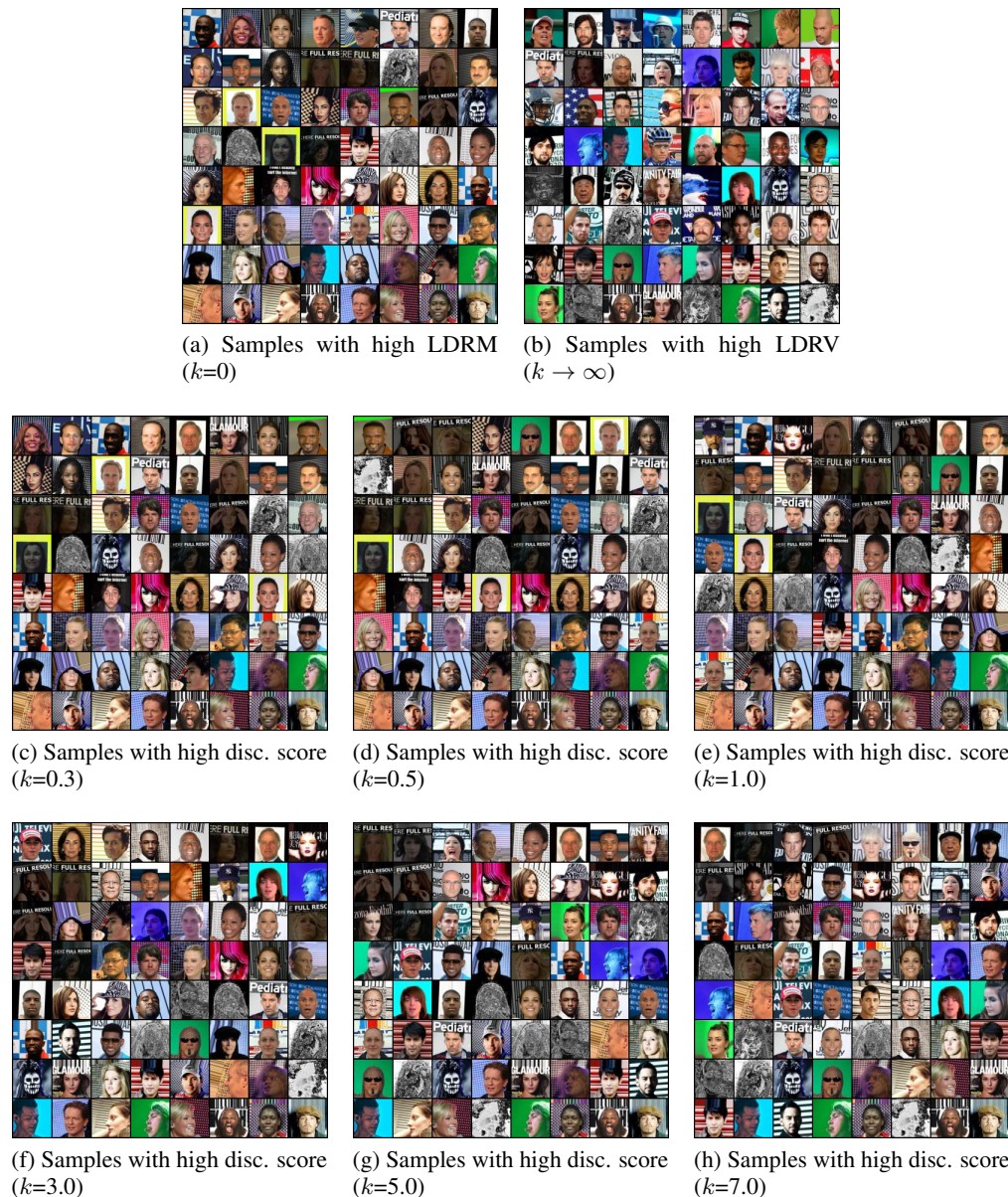

(a) Samples with high LDRM ($k$=0)

(b) Samples with high LDRV ($k \rightarrow \infty$)

(c) Samples with high disc. score ($k$=0.3)

(d) Samples with high disc. score ($k$=0.5)

(e) Samples with high disc. score ($k$=1.0)

(f) Samples with high disc. score ($k$=3.0)

(g) Samples with high disc. score ($k$=5.0)

(h) Samples with high disc. score ($k$=7.0)

Figure A5: CelebA samples with highest discrepancy scores on various $k$

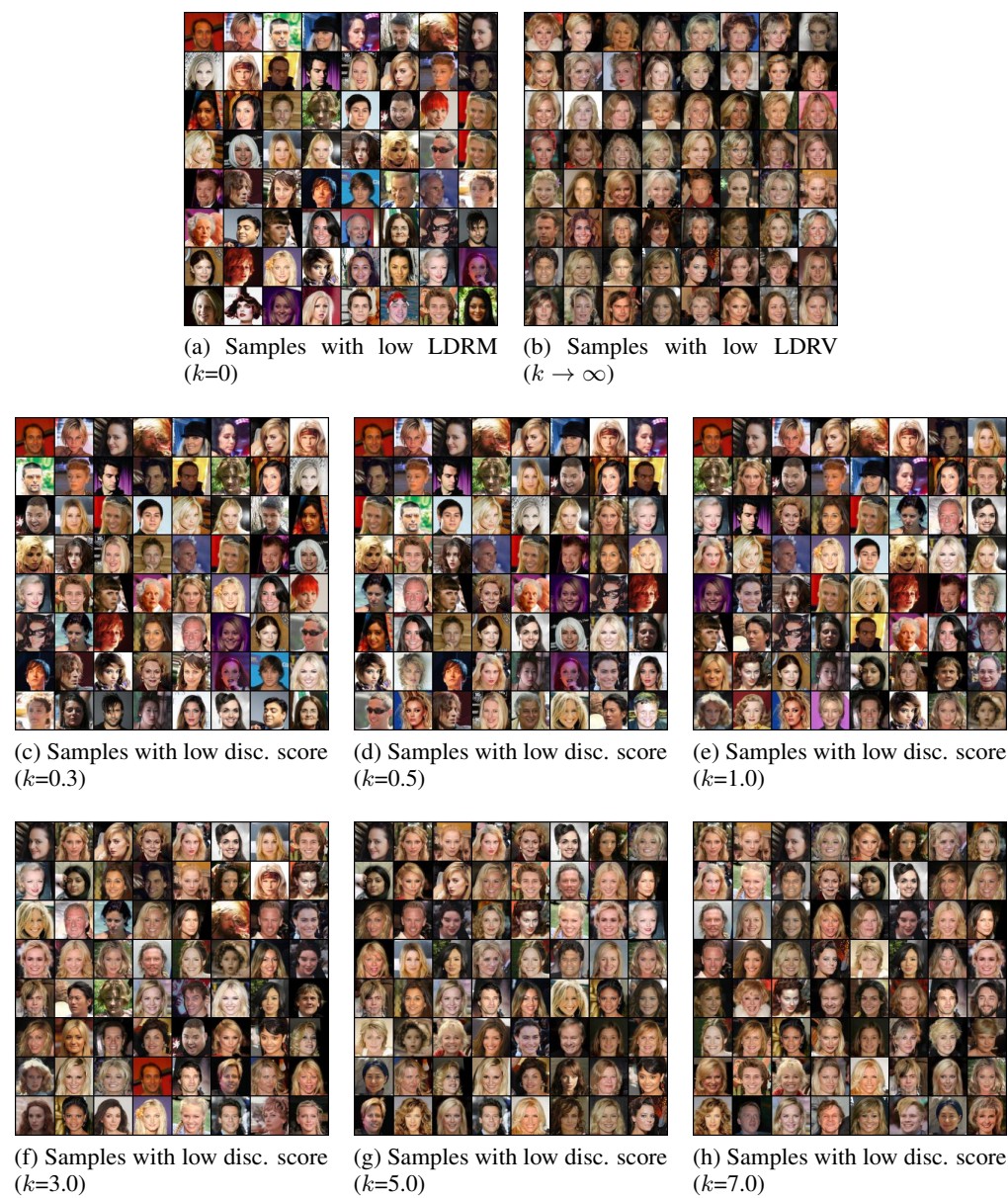

(a) Samples with low LDRM ($k$=0)

(b) Samples with low LDRV ($k \to \infty$)

(c) Samples with low disc. score ($k$=0.3)

(d) Samples with low disc. score ($k$=0.5)

(e) Samples with low disc. score ($k$=1.0)

(f) Samples with low disc. score ($k$=3.0)

(g) Samples with low disc. score ($k$=5.0)

(h) Samples with low disc. score ($k$=7.0)

Figure A6: CelebA samples with lowest discrepancy scores on various $k$

## D Algorithm

Detailed algorithm description for Self-Diagnosing GAN is introduced in Algorithm 1.

---

**Algorithm 1** Self-Diagnosing GAN(Dia-GAN)

---

**Input:** Dataset $\mathcal{D}$, Model $\mathcal{M} = \{D, G, D_{\mathsf{aux}}\}$, Batch size $B$, Numbers of steps for phase 1 and 2 ($t_1$ and $t_2$), step to start recording LDR $t_s$, Number of samples to be generated $N$

**Output:** Set of generated samples $\{g_1, g_2, \cdots, g_N\}$

**Phase 1 - Train and Diagnose**

Initialize $\theta_D^0, \theta_G^0$

**for** $t \leftarrow 1$ **to** $t_1$ **do**

    $\mathcal{D}_B^t \leftarrow \{x_i : x_i \sim \mathrm{Unif}(\mathcal{D})\}$

    $\mathcal{Z}^t \leftarrow \{G(z_j) : z_j \sim p_z(z)\}$

    $\theta_D^t \leftarrow \theta_D^{t-1} + \eta_D \nabla_{\theta_D} V_{\mathsf{D}}(D, G; \mathcal{D}_B^t, \mathcal{Z}^t)$ for $V_D$ in (E.1) (NS loss) or (E.3) (hinge loss)

    $\theta_G^t \leftarrow \theta_G^{t-1} - \eta_G \nabla_{\theta_G} V_{\mathsf{G}}(D, G; \mathcal{Z}^t)$ for $V_G$ in (E.2) (NS loss) or (E.4) (hinge loss).

    **if** $t \geq t_s$ **then**

        Save $\mathsf{LDR}(x_i)_t$ for $i \in \{1, 2, \cdots, |\mathcal{D}|\}$

    **end if**

**end for**

Compute discrepancy score $s(x_i; \{t_s, t_s + 1, \cdots, t_1\})$ for $i \in \{1, 2, \cdots, |\mathcal{D}|\}$. (Eq. (7))

Compute sampling frequency $P_s(i) \propto \mathtt{max\_clip}(\mathtt{min\_clip}(s(x_i; T)))$, for $i \in \{1, 2, \cdots, |\mathcal{D}|\}$.

**Phase 2 - Score-Based Weighted Sampling**

Initialize $\theta_{D_{\mathsf{aux}}}^{t_1} \leftarrow \theta_D^{t_1}$

**for** $t \leftarrow t_1 + 1$ **to** $t_1 + t_2$ **do**

    $\mathcal{D}_B^t \leftarrow \{x_i : x_i \sim P_s(i)\}$

    $\mathcal{D}_B^{t\,'} \leftarrow \{x_i : x_i \sim \mathrm{Unif}(\mathcal{D})\}$

    $\mathcal{Z}^t \leftarrow \{G(z_j) : z_j \sim p_z(z)\}$

    $\theta_D^t \leftarrow \theta_D^{t-1} + \eta_D \nabla_{\theta_D} V_{\mathsf{D}}(D, G; \mathcal{D}_B^t, \mathcal{Z}^t)$

    $\theta_G^t \leftarrow \theta_G^{t-1} - \eta_G \nabla_{\theta_G} V_{\mathsf{G}}(D, G; \mathcal{Z}^t)$

    $\theta_{D_{\mathsf{aux}}}^t \leftarrow \theta_{D_{\mathsf{aux}}}^{t-1} + \eta_{D_{\mathsf{aux}}} \nabla_{\theta_{D_{\mathsf{aux}}}} V_{\mathsf{D}}(D_{\mathsf{aux}}, G; \mathcal{D}_B^{t\,'}, \mathcal{Z}^t)$

**end for**

**Phase 3 - DRS**

$\{g_1, g_2, \cdots, g_N\} \leftarrow \mathsf{DRS}(G; D_{\mathsf{aux}}, N)$

---

**Algorithm complexity**    Compared to the original GAN training, the overhead in time and space from our method is not dominant. For CIFAR-10 dataset, 5 hours 38 minutes were required to train 50k steps of Dia-SNGAN (our method), while 4 hours 51 minutes were needed for the original SNGAN in RTX 3090 GPU. Similarly, for CelebA dataset, 19 hours 53 minutes were required to train 75k steps of Dia-SNGAN (our method), while 17 hours 7 minutes were needed for the original SNGAN with the same GPU. Diagnosing samples in Phase 1 requires additional space for saving LDR values. Phase 2 needs additional auxiliary discriminator training to perform DRS in Phase 3. However, this does not require much overhead since Phase 2 is shorter than Phase 1 and we initialize auxiliary discriminator using the original discriminator trained in Phase 1.

## E Variants of the original GAN loss and description of evaluation metrics

### E.1 Non-saturating GAN loss

We consider a practical training method of GANs, using alternative SGD, to solve $\min_G \max_D V(D, G)$ for $V(D, G) = \mathbb{E}_{x \sim p_{\mathrm{data}}}[\log D(x)] + \mathbb{E}_{z \sim p_z}[\log(1 - D(G(z)))]$. The mini-batches of $B$ samples for the training dataset and fake samples are defined as $\mathcal{D}_B = \{x^{(j)} : x^{(j)} = x_i \text{ where } i \sim P_s(i) \text{ for } j = 1, \ldots, B\}$ and $\mathcal{Z} = \{G(z^{(j)}) : z^{(j)} \sim p_z(z) \text{ for } j = 1, \ldots, B\}$, respectively. Then, the alternative training of GAN updates the discriminator parameter $\theta_D$ and the generator parameter $\theta_G$ by backpropagating the gradient of GAN loss calculated on these mini-batches.

For training, we use the non-saturating variant of the generator loss,

$$V_{\mathsf{D}}(D, G; \mathcal{D}_B, \mathcal{Z}) = \frac{1}{|\mathcal{D}_B|} \sum_{\mathcal{D}_B} \log D(x^{(j)}) + \frac{1}{|\mathcal{Z}|} \sum_{\mathcal{Z}} \log(1 - D(G(z^{(j)}))), \quad \text{(E.1)}$$

$$V_{\mathsf{G}}(D, G; \mathcal{Z}) = -\frac{1}{|\mathcal{Z}|} \sum_{\mathcal{Z}} \log D(G(z^{(j)})). \quad \text{(E.2)}$$

## E.2 Hinge GAN loss

Several types of GANs achieve enhanced performance when the hinge loss [11, 22] is applied instead of the normal non-saturating loss. In Section 5.2, we demonstrate the applicability of our score-based weighted sampling to GANs with hinge loss. The hinge loss is defined as

$$V_{\mathsf{D}}(D, G; \mathcal{D}_B, \mathcal{Z}) = \frac{1}{|\mathcal{D}_B|} \sum_{\mathcal{D}_B} \min(0, -1 + D(x^{(j)})) + \frac{1}{|\mathcal{Z}|} \sum_{\mathcal{Z}} \min(0, -1 - D(G(z^{(j)}))),$$

$$\text{(E.3)}$$

$$V_{\mathsf{G}}(D, G; \mathcal{Z}) = -\frac{1}{|\mathcal{Z}|} \sum_{\mathcal{Z}} D(G(z^{(j)})). \quad \text{(E.4)}$$

## E.3 Description of evaluation metrics

To evaluate the effect of our method on learned model distribution, we use various evaluation metrics including (1) Fréchet Inception Distance (FID) [4], (2) Inception Score (IS) [18], and (3) Precision and Recall (P&R) [9].

- FID measures the distance between the model distribution and the data distribution, approximated by two multidimensional Gaussian distributions in the feature space of InceptionV3 [20] classifier, so it measures the overall fitness of the model distribution to the data distribution, in terms of both the quality (fidelity) and the diversity.

- IS measures the quality of generated samples, in the sense that whether the generated samples can be classified by InceptionV3 classifier into each of the dataset classes.

- Precision is described as the portion of generated samples that fall within the data manifold, which measures the fidelity of generated samples, while recall measures the portion of data instances falling within the manifold of generated samples, which measures the diversity. We follow the definitions of precision and recall by [9], which are described as follows. Let the set of feature vectors of real and generated samples be $\Phi_r$, $\Phi_g$, respectively. Also, let the binary function $f(\phi, \Phi)$ be

$$f(\phi, \Phi) = \begin{cases} 1, & \exists \phi' \in \Phi \text{ s.t. } \|\phi - \phi'\|_2 \leq \|\phi' - \text{NN}_k(\phi', \Phi)\|_2 \\ 0, & \text{otherwise} \end{cases} \quad \text{(E.5)}$$

where $\text{NN}_k(\phi', \Phi)$ denotes the $k-$th nearest feature vector of $\phi'$ in set $\Phi$. Then, precision and recall is defined as:

$$\texttt{precision} = \frac{1}{|\Phi_g|} \sum_{\phi_g \in \Phi_g} f(\phi_g, \Phi_r) \quad \text{(E.6)}$$

$$\texttt{recall} = \frac{1}{|\Phi_r|} \sum_{\phi_r \in \Phi_r} f(\phi_r, \Phi_g) \quad \text{(E.7)}$$

Partial Recall is proposed to measure the recall rate for a subset of dataset. It is defined as the portion of data instances in the subset that fall within the manifold of generated samples. Let us denote a subset of data and the feature space of that subset by $S$ and $\Phi_S$, respectively. Note that $\Phi_S \subset \Phi_r$. Then, the partial recall of the subset $S$ is defined as

$$\texttt{partial\_recall}(S) = \frac{1}{|\Phi_S|} \sum_{\phi_S \in \Phi_S} f(\phi_S, \Phi_g) \quad \text{(E.8)}$$

In addition to these global evaluation metrics, to evaluate whether a subset of dataset is well represented in the model distribution, we consider (4) Reconstruction Error (RE) [24, 27].

- Reconstruction Error (RE) score is calculated by training a convolutional autoencoder (CAE) with generated samples and then calculating the Euclidean distance between each training data and its reconstruction. RE can assess whether $p_g(x)$ covers $p_{\text{data}}(x)$ since CAE is known to have high RE for out-of-distribution samples [24, 27]. RE score for a subset of data is defined as the average RE score of each data instance within the subset. Let us denote the subset of data for which we want to measure RE score by $S$ and the set of generated samples by $F$. The autoencoder output function, trained with samples in $F$, is denoted by $\theta_F(\cdot)$. Then, the RE score of a subset $S$ is defined as

$$\texttt{RE}(S) = \frac{1}{|S|} \sum_{s \in S} \texttt{dist}(\theta_F(s), s) \tag{E.9}$$

where $\texttt{dist}(x, y)$ measures the distance between $x$, $y$ and is defined as the Euclidean distance averaged for all pixels.

For data statistics to calculate FID, we use provided results for CIFAR-10[4] and calculate statistics for CelebA with all the training samples. We compare the statistics of 50,000 generated samples with these data statistics. We use 50,000 generated samples to evaluate IS, and 10,000 data samples and 10,000 generated samples to evaluate P&R. Also, we use the feature layer of the InceptionV3 classifier instead of the feature layer of VGG16 as in [9].

## F    Details of simulation setups

### F.1    Controlled dataset - single-mode Gaussian

We generate 2-D single-mode Gaussian dataset with mean $\mathbf{0}$ and various covariance $\sigma\mathbf{I}$. We use $\sigma \in \{3, 2.5, 2\}$. Minority level (Fig. 1e) 1, 2, 3 stands for $\sigma = 3, 2.5, 2$, respectively. As $\sigma$ decreases, the samples concentrate more near the mean of the Gaussian, and this aligns with the situation that minority rate decreases in the Colored MNIST dataset or the MNIST-FMNIST mixture dataset. The size of the dataset is 10,000. We use a GAN architecture based on the multi-layer perceptron (MLP) with details described in Table A1. We use the batch size of 1024 and Adam optimizer with hyperparameters $\alpha = 0.001, \beta_1 = 0.5, \beta_2 = 0.9$. We train the model for 200 epochs and record LDR for every sample at the end of each epoch while training. We define the major group as the samples within distance two from the origin, and the minor group as the samples outside of distance seven from the origin. To compute Partial Recall, we use data itself as the feature. Fig 1e and Table 1 are the experimental results averaged from 10 random seeds.

### F.2    Controlled dataset - 25 Gaussian dataset

We construct the mixture of 25 Gaussians dataset, each centered at $(c_x, c_y)$ for $c_x, c_y \in \{-2, -1, 0, 1, 2\}/1.414$. Each $(x, y) \in \mathcal{D}$ is sampled from

$$(x, y) = \frac{(d_x, d_y) + (z_x, z_y)}{2.828} \tag{F.1}$$

where $d_x, d_y \in \{-4, -2, 0, 2, 4\}$ and $z_x, z_y \sim N(0, 0.05^2)$. The size of the dataset is 10,000, where 400 samples are sampled from each mixture mode. We use the same GAN architecture as the one used in the single-mode Gaussian experiment (Table A1). We use the batch size of 128 and Adam optimizer with hyperparameters $\alpha = 0.0002, \beta_1 = 0.5, \beta_2 = 0.999$. We train the model for 300 epochs and record LDR for every sample at the end of each epoch while training.

### F.3    Controlled dataset - Colored MNIST & MNIST-FMNIST

We generate Colored MNIST by randomly picking 60,000 samples and separating them into two groups corresponding to each color (red and green) at a given majority rate $\rho$. For the mixture

---
[4]http://bioinf.jku.at/research/ttur/

Table A1: GAN architecture for Single-mode Gaussian and 25 Gaussian dataset

| Generator | | | Discriminator | | |
|---|---|---|---|---|---|
| Layer | Output size | Activation | Layer | Output size | Activation |
| Input $z$ | 2 | | Input $x$ | 2 | |
| FC | 512 | ReLU | FC | 512 | ReLU |
| FC | 512 | ReLU | FC | 512 | ReLU |
| FC | 512 | ReLU | FC | 512 | ReLU |
| FC | 2 | | FC | 1 | Sigmoid |

of MNIST and FMNIST dataset, we randomly pick 60,000 samples in total from MNIST and FMNIST dataset with a given majority rate. For both datasets, minority level (Fig. 1e) 1, 2, 3 stands for the majority rate $\rho = 90\%, 95\%, 99\%$, respectively. We use DCGAN [17] with the detailed architecture described in Table A2 and A3. We use the batch size of 64 and Adam optimizer [8] with hyperparameters $\alpha = 0.0001, \beta_1 = 0.5, \beta_2 = 0.9$. We additionally use the linear learning rate scheduler that decays until the end of the training. All models are trained for 20k steps. For PacGAN [12], we use a packing degree of two. For Inclusive GAN [25], we use Inception feature [18] for the feature space. For GOLD [16] and our method, the phase 1 takes 15k steps, and the phase 2 takes 5k steps. For our method, we record LDR every 100 steps and use the last 50 records for calculating the discrepancy score. We use $k = 3$ for Colored MNIST and $k = 6$ for MNIST-FMNIST.

To evaluate Partial Recall in Fig. 1e, we train convolutional classifier (Table A4) with 60,000 samples (30,000 major samples and 30,000 minor samples) with 20 classes (Major 10 classes + Minor 10 classes) and use output of flatten layer of this network for the feature space. The convolutional classifier is trained for 50 epochs with Adam optimizer [8] with hyperparameters $\alpha = 0.001, \beta_1 = 0.9, \beta_2 = 0.999$ and learning rate scheduler with learning rate decay 0.1 in 42 epoch. To evaluate reconstruction error (RE) in Table 5, we use convolutional autoencoder with the detailed architecture described in Table A5, A6. `nc` in each table states the number of channel. `nc` for Colored MNIST is three and `nc` for MNIST-FMNIST is one. Fig 1e and Table 1, 5 are the experimental results averaged from three random seeds.

Table A2: Generator architecture for Colored MNIST & MNIST-FMNIST

| Generator | | | | | | |
|---|---|---|---|---|---|---|
| Layer | Output size | Kernel size | Stride | Padding | Batch Norm | Activation |
| Input $z$ | 100 | | | | | |
| FC | 384 | - | - | - | X | |
| Reshape | 1×1×384 | - | - | - | - | - |
| Deconv | 4×4×192 | 4×4 | 1 | 0 | O | ReLU |
| Deconv | 8×8×96 | 4×4 | 2 | 1 | O | ReLU |
| Deconv | 16×16×48 | 4×4 | 2 | 1 | O | ReLU |
| Deconv | 32×32×nc | 4×4 | 2 | 1 | X | Tanh |

Table A3: Discriminator architecture for Colored MNIST & MNIST-FMNIST

| Discriminator | | | | | | | |
|---|---|---|---|---|---|---|---|
| Layer | Output size | Kernel size | Stride | Padding | Batch Norm | Dropout | Activation |
| Input $x$ | 32×32×nc | | | | | | |
| Conv | 16×16×16 | 3×3 | 2 | 1 | X | 0.5 | LeakyReLU(0.2) |
| Conv | 16×16×32 | 3×3 | 1 | 1 | O | 0.5 | LeakyReLU(0.2) |
| Conv | 8×8×64 | 3×3 | 2 | 1 | O | 0.5 | LeakyReLU(0.2) |
| Conv | 8×8×128 | 3×3 | 1 | 1 | O | 0.5 | LeakyReLU(0.2) |
| Conv | 4×4×256 | 3×3 | 2 | 1 | O | 0.5 | LeakyReLU(0.2) |
| Conv | 4×4×512 | 3×3 | 1 | 1 | O | 0.5 | LeakyReLU(0.2) |
| Flatten | - | - | - | - | - | - | - |
| FC | 1 | - | - | - | X | | Sigmoid |

Table A4: Classifier architecture for measuring Partial Recall of Colored MNIST & MNIST-FMNIST

| | | Classifier | | | | |
|---|---|---|---|---|---|---|
| Layer | Output size | Kernel size | Stride | Padding | Batch Norm | Activation |
| Input $x$ | 32×32×nc | | | | | |
| Conv | 32×32×16 | 7×7 | 1 | 3 | O | ReLU |
| Conv | 32×32×32 | 7×7 | 1 | 3 | O | ReLU |
| Conv | 32×32×64 | 7×7 | 1 | 3 | O | ReLU |
| Conv | 32×32×128 | 7×7 | 1 | 3 | O | ReLU |
| AvgPool | 1×1×128 | - | - | - | - | - |
| Flatten | - | - | - | - | - | - |
| FC | 20 | - | - | - | X | Softmax |

Table A5: Encoder architecture for measuring Reconstruction Error (RE) score

| | | Encoder | | | | |
|---|---|---|---|---|---|---|
| Layer | Output size | Kernel size | Stride | Padding | Batch Norm | Activation |
| Input $x$ | 32×32×nc | | | | | |
| Conv | 16×16×64 | 3×3 | 2 | 1 | O | ReLU |
| Conv | 8×8×128 | 3×3 | 2 | 1 | O | ReLU |
| Conv | 4×4×256 | 3×3 | 2 | 1 | O | ReLU |
| Flatten | - | - | - | - | - | - |
| FC | 256 | - | - | - | X | Tanh |

Table A6: Decoder architecture for measuring Reconstruction Error (RE) score

| | | Decoder | | | | | |
|---|---|---|---|---|---|---|---|
| Layer | Output size | Kernel size | Stride | Padding | Output padding | Batch Norm | Activation |
| Input $z$ | 256 | | | | | | |
| FC | (4×4×256) | - | - | - | - | O | ReLU |
| Reshape | 4×4×256 | - | - | - | - | - | - |
| Deconv | 8×8×128 | 3×3 | 2 | 1 | 1 | O | ReLU |
| Deconv | 16×16×64 | 3×3 | 2 | 1 | 1 | O | ReLU |
| Deconv | 32×32×nc | 3×3 | 2 | 1 | 1 | X | Tanh |

## F.4   Real dataset - CIFAR-10 and CelebA

We evaluate our method with two types of GANs: SNGAN [15] and SSGAN [23] [5]. Following [15], we use the residual network architecture proposed in ResNet [3] for all GAN variants. Our experimental code is based on the GAN research library Mimicry [10]. We use batch size of 64 and Adam optimizer [8] with hyperparameters $\alpha = 0.0002, \beta_1 = 0, \beta_2 = 0.9$. The learning rate is set to decay linearly with the training steps. Table 2 and 4 are the experimental results averaged from three random seeds.

## F.5   Real dataset - FFHQ

We test the scalability of our method on the large-scale model. Specifically, we train StyleGAN2 [7] on FFHQ 256x256 [6] dataset. We follow most of the techniques used in the original StyleGAN2 [7]. We use leaky ReLU activation with $\alpha = 0.2$, bilinear filtering [26] in all up/downsampling layers, minibatch standard deviation layer at the end of the discriminator [5], exponential moving average of generator weights [5] and style mixing regularization [6]. For the discriminator, we use $r_1 = 0.1$ for the weight of $R_1$ regularizer [14] and apply the lazy regularization [7] every 16 steps. For the generator, we apply path length regularization [7] with weight of 2 and batch size reducing factor of 2 and also apply the lazy regularization every 4 steps. We use the batch size of 16 and Adam optimizer [8] with the hyperparameters $\alpha = 0.0016, \beta_1 = 0, \beta_2 = 0.991$. In total, we train for 250k

---

[5]When we train SSGAN with the Top-k method, we only consider top-$k$ samples for the GAN tasks, while we use full (not top-$k$) samples for the self-supervised tasks.

where the phase 1 takes 200k steps and phase 2 takes the remaining 50k steps. We record the LDR values every 100 steps for the last 5k steps of phase 1 (195k $\sim$ 200k). For the discrepancy score, we use $k = 3.0$. Table 3 shows the experimental results averaged from two random seeds.

## F.6 Details on CelebA minor attribute analysis

To analyze the CelebA minor attribute, we use the meta-information provided by CelebA [13]. We use a pre-trained VGG16 network to train attribute classifiers for each attribute. Except for the last three fully connected layers, we fix the parameters of the pre-trained VGG16 network and change the output size of last layer to two. We train only the last three layers (classifier layers) of the VGG16 network for 10 epochs with batch size 128 and SGD optimizer with a learning rate of 0.001 and momentum of 0.9. We select the minor attributes with accuracy above 95% for train and test datasets. We count the occurrence of minor attributes using the trained classifier. Table 6 is the experimental results averaged from three random seeds.

## F.7 Hyperparameter search for discrepancy score

The hyperparmeter $k$ for discrepancy score (7) is chosen from $k = 0.3, 0.5, 1.0, 3.0, 5.0, 7.0$ to achieve the best FID score among the candidates for each dataset at SNGAN, and the value of $k$ is fixed across the GAN variants. See Table A7 for details. As we can see in Table A7, an appropriate choice of $k$ can be different depending on the dataset. These are results averaged from three random trials.

Table A7: FID for Dia-GAN with various $k$

| $k$ | 0.3 | 0.5 | 1.0 | 3.0 | 5.0 | 7.0 |
|---|---|---|---|---|---|---|
| FID for CIFAR-10 | **19.23** | 19.47 | 20.58 | 23.45 | 24.44 | 20.43 |
| FID for CelebA | 7.27 | 6.91 | 6.52 | 6.73 | **6.37** | 6.41 |

## F.8 Hyperparameter choice for training steps

We choose the training steps for Phase 1 of our algorithm as 80% of total steps to make sure that the discriminator is trained enough. However, experiments with the different training step choices shown in Table A8 imply our method's robustness on the parameter choice.

Table A8: FID for Dia-GAN with different phase 1 steps (% of total steps).

| | Baseline | 20% | 40% | 60% | 80% |
|---|---|---|---|---|---|
| FID for CIFAR-10 | 26.90±0.90 | 17.56±1.03 | **16.72±0.74** | 18.65±0.94 | 19.66±0.42 |
| FID for CelebA | 7.12±0.27 | **6.69±0.33** | 6.90±0.66 | 6.86±0.77 | 6.70±0.69 |

When we take the longer total training steps as 100k steps for SNGAN on CIFAR-10 and CelebA, we find similar trends as we use 50k steps for total training steps. See Table A9 for details. The overall FID gets better when the model is trained longer, but our method still gives an improvement in term of FID, Inception score and recall. In addition, we want to point out that our method can offer an efficient way of training, as our method requires much fewer steps to achieve FID better that the best FID of the Vanilla GAN.

## F.9 Necessity of combining LDRM and LDRV

Our discrepancy score is the weighted sum of two metrics, balancing the effects of two terms. To check the effects of combining two metrics, we train SNGAN on CIFAR-10 and CelebA using only LDRM or LDRV metric. We use clipped LDRM or clipped LDRV value as we applied to the discrepancy score. As shown in Table A10, average FID of using only one metric cannot achieves average FID of using discrepancy score. This implies the importance of incorporating LDRV over LDRM and the effect of proper balancing of both metrics.

Table A9: FID for SNGAN and Dia-SNGAN with different total training steps.

| | CIFAR-10 | | CelebA | | |
|---|---|---|---|---|---|
| | FID ↓ | IS ↑ | FID ↓ | P ↑ | R ↑ |
| SNGAN (50k/ 75k) | 26.90±0.90 | 7.36±0.08 | 7.12±0.27 | **0.68**±**0.00** | 0.44±0.01 |
| Dia-SNGAN (50k/ 75k) | **19.66**±**0.42** | **7.95**±**0.09** | **6.70**±**0.69** | 0.64±0.02 | **0.48**±**0.02** |
| SNGAN (100k) | 22.43±0.92 | 7.59±0.06 | 6.83±0.46 | **0.68**±**0.00** | 0.45±0.02 |
| Dia-SNGAN (100k) | **16.49**±**1.05** | **8.10**±**0.14** | **6.57**±**0.70** | 0.63±0.01 | **0.49**±**0.01** |

Table A10: FID for GAN using weighted sampling with LDRM and LDRV.

| | Baseline | LDRM | LDRV | Dia-GAN(Ours) |
|---|---|---|---|---|
| FID for CIFAR-10 | 26.90±0.90 | 19.80±0.47 | 20.08±0.67 | **19.66**±**0.42** |
| FID for CelebA | 7.12±0.27 | 7.46±0.57 | 7.08±0.75 | **6.70**±**0.69** |

## F.10 Details on Discriminator Rejection Sampling (DRS)

In this subsection, we introduce the practical scheme of DRS by briefly referring to original DRS paper and explain our hyperparameter uses for DRS algorithm. Discriminator Rejection Sampling [1] accepts the fake sample $x$ with probability $p_{\mathsf{data}}(x)/Mp_g(x)$ where $M = \max_x (p_{\mathsf{data}}(x)/p_g(x))$. If we let $\mathsf{LDR}_M = \log M$, then acceptance probability for $x$, denoted by $p_{\mathrm{accept}}(x)$, would be

$$p_{\mathrm{accept}}(x) = e^{\mathsf{LDR}(x)-\mathsf{LDR}_M}. \tag{F.2}$$

To deal with low acceptance probabilities and numerical stability issue, Azadi et al. [1] instead proposed to compute $F(x)$ which satisfies

$$p_{\mathrm{accept}}(x) = \frac{1}{1+e^{-F(x)}}. \tag{F.3}$$

Equivalently,

$$F(x) = \mathsf{LDR}(x) - \mathsf{LDR}_M - \log(1 - e^{\mathsf{LDR}(x)-\mathsf{LDR}_M}). \tag{F.4}$$

Practically, in DRS algorithm we compute

$$\hat{F}(x) = \mathsf{LDR}(x) - \mathsf{LDR}_M - \log(1 - e^{\mathsf{LDR}(x)-\mathsf{LDR}_M-\epsilon}) - \gamma, \tag{F.5}$$

where $\epsilon$ is a constant for preventing overflow and $\gamma$ is a hyperparameter for controlling the acceptance probability. For applying DRS with auxiliary discriminator in our algorithm, we used $\epsilon = 10^{-6}$ and let $\gamma$ be 80% percentile of $\hat{F}(x)$. $\mathsf{LDR}_M$ is initially estimated with $256 \times 50 = 12800$ samples by finding the maximum LDR value among those samples. $\mathsf{LDR}_M$ is updated during sampling phase whenever a bigger one is found.

# G  Effect of our method in sample generation for CIFAR-10 & CelebA

## G.1  Visualized effect of weighted sampling

In Fig. A7 (CIFAR-10) and A8 (CelebA), we compare the generated samples with and without our sampling method, which emphasizes underrepresented samples having high discrepancy scores. We also visualize the effect of our weighted sampling by showing some examples of generated samples $G(z)$ with the same $z$ between original GAN and our method in Fig. A9. In Fig. A9, we show some examples of CelebA images with minor features such as accessories including glasses or hats appeared by our weighted sampling, and also images having unique backgrounds (e.g. with some letters in the background) with our method. These minor features are often underrepresented in sample generation of original GANs, while our weighted sampling effectively helps the model learn such minor features by detecting and emphasizing underrepresented samples.

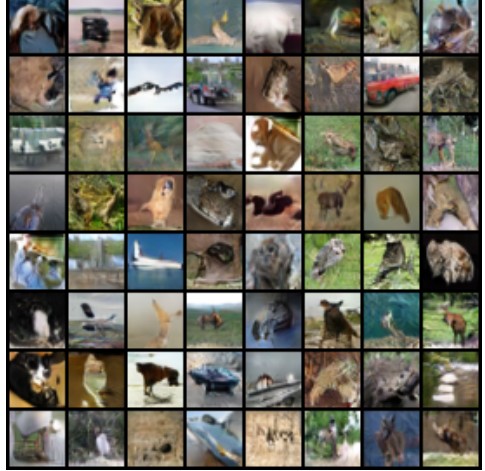 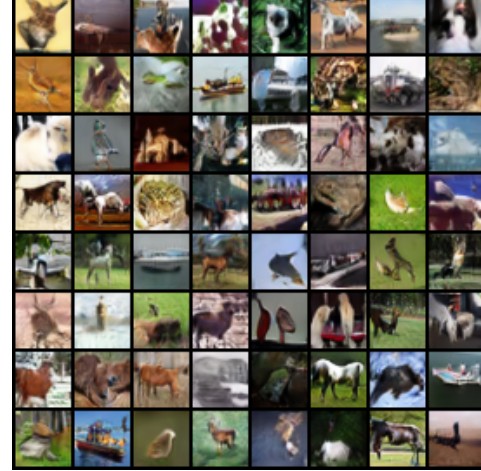

(a) Generated samples with original sampling method

(b) Generated samples with weighted sampling method (ours)

Figure A7: Example of generated samples with (a) original sampling and with (b) weighted sampling (CIFAR-10)

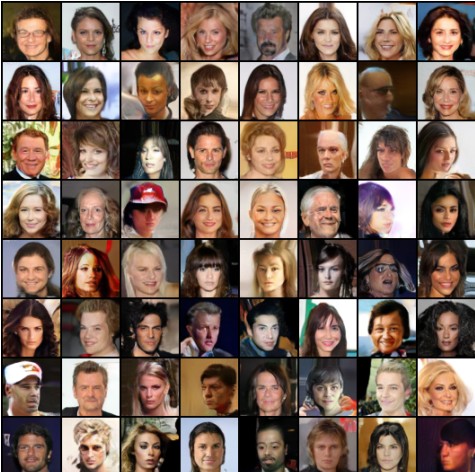 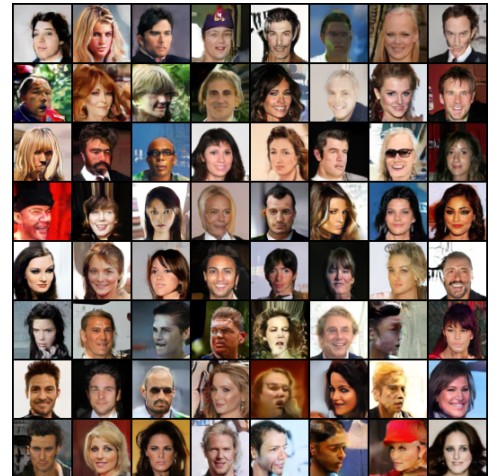

(a) Generated samples with original sampling method

(b) Generated samples with weighted sampling method (ours)

Figure A8: Example of generated samples with (a) original sampling and with (b) weighted sampling (CelebA)

## G.2    Quantitative comparison of generated samples

To verify that our method encourages model to generate underrepresented samples (having high discrepancy scores) for CIFAR-10 and CelebA, we evaluate 'PFID (Partial FID)'. Original FID is calculated by comparing the feature statistics of all training data and randomly sampled generated samples, but PFID is calculated by the difference between the feature statistics of the specific subset of training data and generated samples. We evaluate the High PFID, the PFID of 5,000 training samples having the highest discrepancy scores and the Low PFID, the PFID of 5,000 training samples having the lowest discrepancy scores. In both PFID calculations, we use 50,000 generated samples. The results are summarized in Table A11 (averaged over three trials), where the PFID values are calculated for SNGAN.

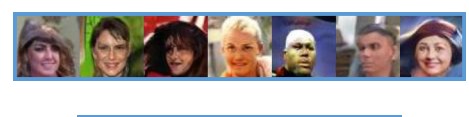
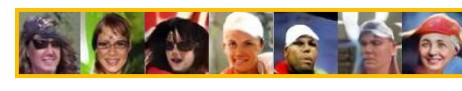

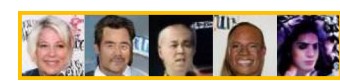

(a) Generated samples with original sampling     (b) Generated samples with weighted sampling

Figure A9: Comparison of generated samples with (a) original sampling and with (b) weighted sampling for CelebA dataset. Examples of generated images with minor features such as accessories including glasses or hats appeared by our method (1st row), or with more unique backgrounds (e.g. with some letters in the background) (2nd row).

This result shows the effectiveness of our method in two aspects. First, the Low PFID is significantly lower than the High PFID, which means that our discrepancy score successfully detects underrepresented samples. Another aspect is that after weighted sampling, the High PFID decreases significantly, implying the effectiveness of our method on promoting the consideration of high-scoring, underrepresented samples in GAN training.

Table A11: Partial FID for SNGAN

|  |  | Baseline | Dia-GAN (ours) |
|---|---|---|---|
| CIFAR-10 | High PFID | $94.64_{\pm2.93}$ | $77.28_{\pm3.76}$ |
|  | Low PFID | $22.43_{\pm0.68}$ | $33.98_{\pm1.77}$ |
| CelebA | High PFID | $50.25_{\pm3.24}$ | $42.33_{\pm3.36}$ |
|  | Low PFID | $17.25_{\pm1.35}$ | $23.17_{\pm3.29}$ |

## H  Effect of our method in capturing semantic features for FFHQ

To ensure that the ability of our method in capturing semantic features also applies to high-resolution datasets, we consider the FFHQ dataset and classify the race on the FFHQ dataset using the DeepFace architecture [21]. This architecture classifies the images as Asian, Black, Indian, Latino hispanic, Middle eastern, and White. For the FFHQ dataset, Black, Indian, and Middle eastern represent the minority taking less than 5% of the FFHQ dataset. See Table A12 for details.

We compare the occurrence rate and partial recall for these minor races after training with vanilla StyleGAN2 and Dia-GAN, respectively. The results are shown in Table A13.

Table A12: Ratio(%) of race on the FFHQ dataset classified by the DeepFace architecture [21].

| Race | Asian | Black | Indian | Latino hispanic | Middle eastern | White |
|---|---|---|---|---|---|---|
| Ratio(%) | 19.38 | 4.80 | 2.08 | 10.83 | 4.04 | 58.87 |

Table A13: FFHQ minor attribute analysis. O stands for the occurrence of minor attributes among the generated samples in percentage (%) and R stands for the Partial Recall.

|  | Vanilla | | Dia-GAN | |
|---|---|---|---|---|
|  | O ↑ | R ↑ | O ↑ | R ↑ |
| Black (4.80%) | $\mathbf{3.00}_{\pm\mathbf{0.13}}$ | $0.27_{\pm0.03}$ | $2.99_{\pm0.17}$ | $\mathbf{0.30}_{\pm\mathbf{0.01}}$ |
| Indian (2.08%) | $0.81_{\pm0.19}$ | $0.26_{\pm0.03}$ | $\mathbf{1.16}_{\pm\mathbf{0.03}}$ | $\mathbf{0.30}_{\pm\mathbf{0.01}}$ |
| Middle eastern (4.04%) | $3.18_{\pm0.20}$ | $0.27_{\pm0.04}$ | $\mathbf{3.49}_{\pm\mathbf{0.61}}$ | $\mathbf{0.31}_{\pm\mathbf{0.00}}$ |

Similar to the results for the CelebA dataset in Section 5.3, the occurrence rate and partial recall for minor races in the FFHQ dataset are improved with our method, especially for Indian and Middle-eastern image samples.

In conclusion, this evidence demonstrates that our method successfully captures semantically meaningful minor attributes and emphasizes them during the training, resulting in a diverse generation of minor samples across low- to high-resolution datasets.

# I Examples of generated samples for MNIST-FMNIST

We show randomly generated samples of various GANs trained on MNIST-FMNIST with different majority (MNIST) rate $\rho \in \{90, 95, 99\}\%$ in Fig. A10. Our method is the only method that recovers the minor (FMNIST) features when the rate is 99%. Moreover, the reconstruction error (RE) scores reported in Table 5 demonstrate that our method improves the quality of generated samples with minor features, resulting in better RE score of the green training samples. Results indicate the effectiveness of Dia-GAN in improving the quality of generated samples with underrepresented features.

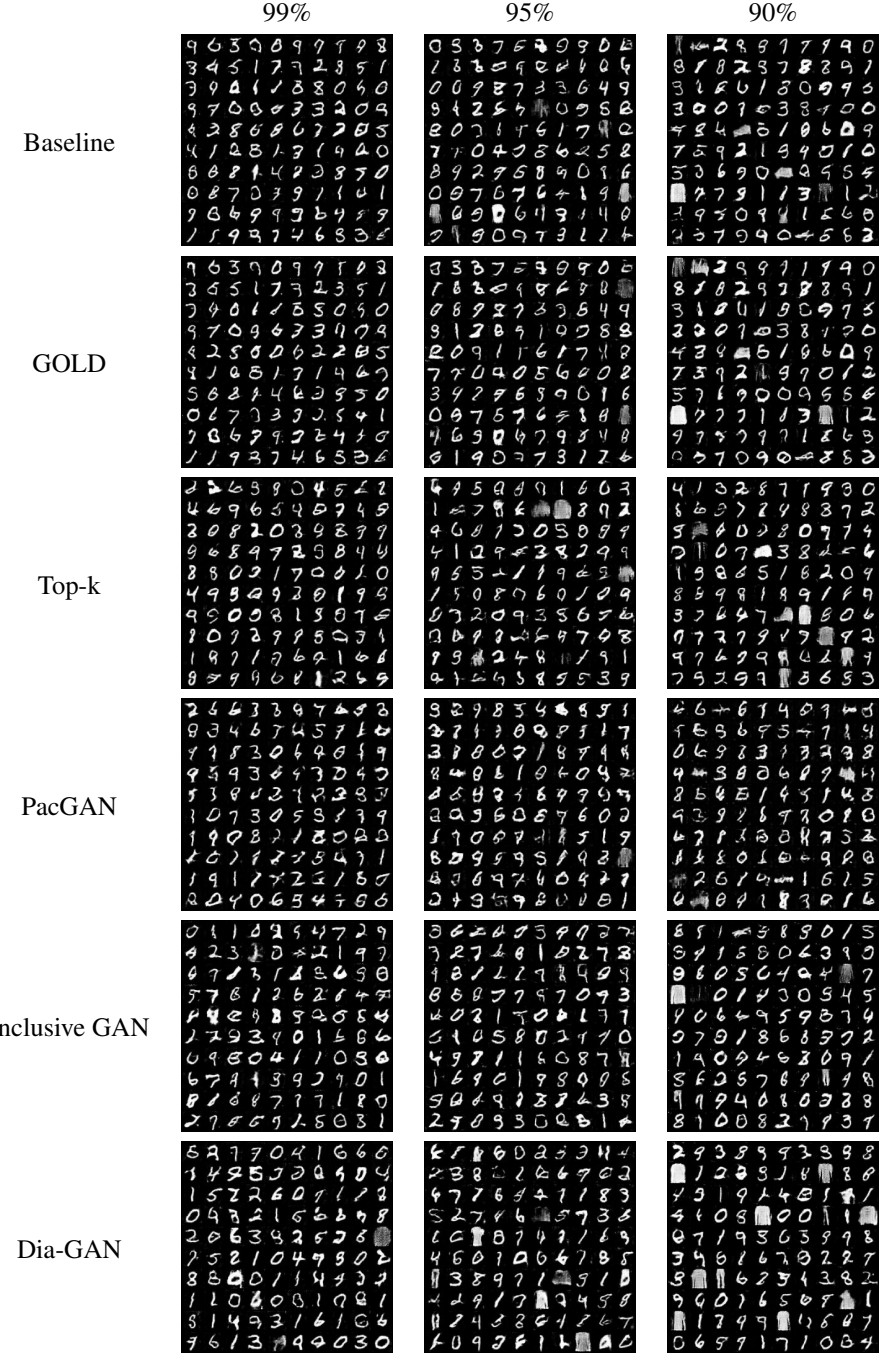

Figure A10: MNIST-FMNIST generated samples of various GANs on different majority rate.

# J   Full experimental results with standard deviation

In this section, we show the detailed results for the tables in the main document, which are reported with only mean values due to the space limitation, with the standard deviation.

Table A14: (Details of Table 1) LDRV of major/minor groups on various datasets with majority rate 90%.

| Group | Gaussian ($\sigma$=3.0) | Colored MNIST | MNIST-FMNIST |
|---|---|---|---|
| Major | $0.001_{\pm 0.000}$ | $0.077_{\pm 0.018}$ | $0.082_{\pm 0.022}$ |
| Minor | $0.098_{\pm 0.009}$ | $0.186_{\pm 0.057}$ | $0.115_{\pm 0.021}$ |

Table A15: (Details of Table 2) Comparison of diverse sampling/weighting techniques for CIFAR-10 image generation.

| Methods | SNGAN | | SSGAN | |
|---|---|---|---|---|
| | **FID** $\downarrow$ | **IS** $\uparrow$ | **FID** $\downarrow$ | **IS** $\uparrow$ |
| Vanilla | $26.90_{\pm 0.90}$ | $7.36_{\pm 0.08}$ | $22.01_{\pm 0.99}$ | $7.65_{\pm 0.10}$ |
| DRS [1] | $24.54_{\pm 0.80}$ | $7.57_{\pm 0.05}$ | $20.51_{\pm 1.01}$ | $7.77_{\pm 0.09}$ |
| GOLD [16] | $28.86_{\pm 0.92}$ | $7.21_{\pm 0.09}$ | $21.90_{\pm 0.90}$ | $7.57_{\pm 0.09}$ |
| GOLD + DRS [1] | $24.65_{\pm 0.86}$ | $7.53_{\pm 0.09}$ | $19.36_{\pm 0.45}$ | $7.79_{\pm 0.04}$ |
| Top-k [19] | $24.45_{\pm 0.60}$ | $7.60_{\pm 0.06}$ | $20.01_{\pm 1.23}$ | $7.78_{\pm 0.08}$ |
| Top-k + DRS [1] | $23.92_{\pm 0.69}$ | $7.70_{\pm 0.09}$ | $20.09_{\pm 0.98}$ | $7.88_{\pm 0.10}$ |
| **Dia-GAN** | $\mathbf{19.66_{\pm 0.42}}$ | $\mathbf{7.95_{\pm 0.09}}$ | $\mathbf{16.31_{\pm 0.53}}$ | $\mathbf{8.14_{\pm 0.06}}$ |

Table A16: (Details of Table 2) Comparison of diverse sampling/weighting techniques for CelebA image generation.

| Methods | SNGAN | | | SSGAN | | |
|---|---|---|---|---|---|---|
| | **FID** $\downarrow$ | **Prec.** $\uparrow$ | **Rec.** $\uparrow$ | **FID** $\downarrow$ | **Prec.** $\uparrow$ | **Rec.** $\uparrow$ |
| Vanilla | $7.12_{\pm 0.27}$ | $0.68_{\pm 0.00}$ | $0.44_{\pm 0.01}$ | $7.19_{\pm 0.18}$ | $0.68_{\pm 0.01}$ | $0.44_{\pm 0.02}$ |
| DRS [1] | $7.04_{\pm 0.31}$ | $0.68_{\pm 0.01}$ | $0.44_{\pm 0.01}$ | $7.08_{\pm 0.23}$ | $0.68_{\pm 0.01}$ | $0.45_{\pm 0.01}$ |
| GOLD [16] | $7.31_{\pm 0.67}$ | $\mathbf{0.69_{\pm 0.00}}$ | $0.44_{\pm 0.02}$ | $7.46_{\pm 0.31}$ | $0.68_{\pm 0.00}$ | $0.43_{\pm 0.00}$ |
| GOLD + DRS [1] | $6.97_{\pm 0.64}$ | $0.68_{\pm 0.01}$ | $0.44_{\pm 0.01}$ | $7.15_{\pm 0.37}$ | $0.67_{\pm 0.01}$ | $0.45_{\pm 0.01}$ |
| Top-k [19] | $7.35_{\pm 0.20}$ | $0.67_{\pm 0.00}$ | $0.44_{\pm 0.01}$ | $7.23_{\pm 0.18}$ | $0.67_{\pm 0.00}$ | $0.45_{\pm 0.01}$ |
| Top-k + DRS [1] | $7.35_{\pm 0.18}$ | $0.68_{\pm 0.00}$ | $0.44_{\pm 0.00}$ | $7.16_{\pm 0.25}$ | $\mathbf{0.68_{\pm 0.00}}$ | $0.45_{\pm 0.00}$ |
| **Dia-GAN** | $\mathbf{6.70_{\pm 0.69}}$ | $0.64_{\pm 0.02}$ | $\mathbf{0.48_{\pm 0.02}}$ | $\mathbf{6.88_{\pm 0.58}}$ | $0.66_{\pm 0.02}$ | $\mathbf{0.46_{\pm 0.02}}$ |

Table A17: (Details of Table 3) StyleGAN2 on FFHQ 256x256.

| | FID ↓ | P ↑ | R ↑ |
|---|---|---|---|
| StyleGAN2 | $14.07_{\pm3.07}$ | $\mathbf{0.72}_{\pm\mathbf{0.02}}$ | $0.27_{\pm0.03}$ |
| GOLD | $15.53_{\pm4.14}$ | $0.69_{\pm0.00}$ | $0.29_{\pm0.02}$ |
| **Dia-StyleGAN2** | $\mathbf{11.89}_{\pm\mathbf{0.21}}$ | $0.69_{\pm0.01}$ | $\mathbf{0.30}_{\pm\mathbf{0.01}}$ |

Table A18: (Details of Table 4) HingeGAN on CIFAR-10 and CelebA.

| | CIFAR-10 | | CelebA |
|---|---|---|---|
| | FID ↓ | IS ↑ | FID ↓ |
| HingeGAN | $21.99_{\pm1.73}$ | $7.67_{\pm0.16}$ | $6.66_{\pm0.06}$ |
| **Dia-HingeGAN** | $\mathbf{18.74}_{\pm\mathbf{1.79}}$ | $\mathbf{8.02}_{\pm\mathbf{0.14}}$ | $\mathbf{5.98}_{\pm\mathbf{0.15}}$ |

Table A19: (Details of Table 5) Reconstruction Error (RE) score of green (minor) training samples in Colored MNIST on different majority rate $\rho$.

| Dataset | Colored MNIST | | |
|---|---|---|---|
| Majority rate $\rho$ | 99% | 95% | 90% |
| Vanilla | $0.838_{\pm0.033}$ | $0.236_{\pm0.037}$ | $0.218_{\pm0.058}$ |
| GOLD [16] | $0.813_{\pm0.002}$ | $0.297_{\pm0.146}$ | $0.200_{\pm0.022}$ |
| Top-k [19] | $0.831_{\pm0.022}$ | $0.210_{\pm0.012}$ | $0.223_{\pm0.015}$ |
| PacGAN [12] | $0.810_{\pm0.001}$ | $0.244_{\pm0.049}$ | $0.233_{\pm0.052}$ |
| Inclusive GAN [25] | $0.812_{\pm0.001}$ | $0.274_{\pm0.060}$ | $0.216_{\pm0.024}$ |
| **Dia-GAN** | $\mathbf{0.224}_{\pm\mathbf{0.020}}$ | $\mathbf{0.204}_{\pm\mathbf{0.018}}$ | $\mathbf{0.197}_{\pm\mathbf{0.026}}$ |

Table A20: (Details of Table 5) Reconstruction Error (RE) score of FMNIST samples (minor) in a mixture of MNIST and FMNIST on different majority rate $\rho$.

| Dataset | MNIST-FMNIST | | |
|---|---|---|---|
| Majority rate $\rho$ | 99% | 95% | 90% |
| Vanilla | $0.290_{\pm0.019}$ | $0.227_{\pm0.001}$ | $0.215_{\pm0.010}$ |
| GOLD [16] | $0.296_{\pm0.008}$ | $0.241_{\pm0.005}$ | $0.218_{\pm0.004}$ |
| Top-k [19] | $0.281_{\pm0.012}$ | $0.232_{\pm0.006}$ | $0.221_{\pm0.007}$ |
| PacGAN [12] | $0.313_{\pm0.026}$ | $0.251_{\pm0.013}$ | $0.225_{\pm0.007}$ |
| Inclusive GAN [25] | $0.283_{\pm0.012}$ | $0.230_{\pm0.015}$ | $0.220_{\pm0.011}$ |
| **Dia-GAN** | $\mathbf{0.264}_{\pm\mathbf{0.007}}$ | $\mathbf{0.219}_{\pm\mathbf{0.016}}$ | $\mathbf{0.206}_{\pm\mathbf{0.002}}$ |

Table A21: (Details on Table 6) CelebA minor attribute analysis. Mean of LDRV and mean of the discrepancy score of CelebA samples with (W/) or without (W/O) minor attributes.

| Method | LDRV | | Discrepancy | |
|---|---|---|---|---|
| | W/ | W/O | W/ | W/O |
| Bald (2.244%) | $\mathbf{0.271}_{\pm\mathbf{0.050}}$ | $0.184_{\pm0.035}$ | $\mathbf{2.938}_{\pm\mathbf{0.183}}$ | $2.221_{\pm0.183}$ |
| Double Chin (4.669%) | $\mathbf{0.219}_{\pm\mathbf{0.040}}$ | $0.184_{\pm0.035}$ | $\mathbf{2.525}_{\pm\mathbf{0.188}}$ | $2.224_{\pm0.183}$ |
| Eyeglasses (6.512%) | $\mathbf{0.254}_{\pm\mathbf{0.048}}$ | $0.181_{\pm0.035}$ | $\mathbf{2.783}_{\pm\mathbf{0.202}}$ | $2.200_{\pm0.182}$ |
| Gray Hair (4.195%) | $\mathbf{0.211}_{\pm\mathbf{0.037}}$ | $0.185_{\pm0.035}$ | $\mathbf{2.450}_{\pm\mathbf{0.173}}$ | $2.228_{\pm0.184}$ |
| Mustache (4.155%) | $\mathbf{0.242}_{\pm\mathbf{0.047}}$ | $0.183_{\pm0.035}$ | $\mathbf{2.699}_{\pm\mathbf{0.218}}$ | $2.218_{\pm0.182}$ |
| Pale Skin (4.295%) | $\mathbf{0.190}_{\pm\mathbf{0.032}}$ | $0.186_{\pm0.036}$ | $\mathbf{2.240}_{\pm\mathbf{0.156}}$ | $2.238_{\pm0.184}$ |
| Wearing Hat (4.846%) | $\mathbf{0.357}_{\pm\mathbf{0.072}}$ | $0.177_{\pm0.034}$ | $\mathbf{3.651}_{\pm\mathbf{0.297}}$ | $2.164_{\pm0.178}$ |

Table A22: (Details of Table 6) CelebA minor attribute analysis. O stands for the occurrence of minor attributes among the generated samples in percentage (%) and R stands for the Partial Recall.

| Method | Vanilla | | Dia-GAN | |
|---|---|---|---|---|
| | O ↑ | R ↑ | O ↑ | R ↑ |
| Bald (2.244%) | $0.678_{\pm 0.164}$ | $0.353_{\pm 0.014}$ | $\mathbf{0.836}_{\pm \mathbf{0.089}}$ | $\mathbf{0.393}_{\pm \mathbf{0.012}}$ |
| Double Chin (4.669%) | $0.440_{\pm 0.090}$ | $0.411_{\pm 0.015}$ | $\mathbf{0.522}_{\pm \mathbf{0.090}}$ | $\mathbf{0.461}_{\pm \mathbf{0.003}}$ |
| Eyeglasses (6.512%) | $3.300_{\pm 0.044}$ | $0.400_{\pm 0.019}$ | $\mathbf{4.053}_{\pm \mathbf{0.282}}$ | $\mathbf{0.449}_{\pm \mathbf{0.008}}$ |
| Gray Hair (4.195%) | $2.273_{\pm 0.335}$ | $0.402_{\pm 0.016}$ | $\mathbf{2.369}_{\pm \mathbf{0.087}}$ | $\mathbf{0.436}_{\pm \mathbf{0.013}}$ |
| Mustache (4.155%) | $0.157_{\pm 0.027}$ | $0.391_{\pm 0.012}$ | $\mathbf{0.228}_{\pm \mathbf{0.009}}$ | $\mathbf{0.433}_{\pm \mathbf{0.008}}$ |
| Pale Skin (4.295%) | $0.346_{\pm 0.014}$ | $0.380_{\pm 0.013}$ | $\mathbf{0.453}_{\pm \mathbf{0.004}}$ | $\mathbf{0.427}_{\pm \mathbf{0.025}}$ |
| Wearing Hat (4.846%) | $2.307_{\pm 0.055}$ | $0.380_{\pm 0.007}$ | $\mathbf{3.595}_{\pm \mathbf{0.655}}$ | $\mathbf{0.408}_{\pm \mathbf{0.020}}$ |