# OpenReview forum: "Self-Diagnosing GAN: Diagnosing Underrepresented Samples in Generative Adversarial Networks"
_NeurIPS.cc/2021/Conference — NeurIPS 2021 Poster_

### Official Review · Reviewer_bY4D · 2021-07-10

**Rating:** 6
**Confidence:** 4

**Summary:**

This paper proposes a sample reweighing scheme for GANs that emphasizes underrepresented samples in the training data. Log-density ratio statistics from the discriminator are used to reweigh the sample probability of data during training, ultimately resulting in more diverse generations.

**Limitations And Societal Impact:**

Authors mention positive impact of making sure that machine learning models include minorities.

**Main Review:**

Strengths:
- As far as I can tell, this is the first work to propose reweighing real samples for unconditional GAN training. Previous work has reweighed fake samples or performed instance selection on real samples (which could be seen as a binary form of reweighing), but none have used the discriminator to inform sampling probabilities of real data.
- Problem is motivated both empirically and theoretically.
- Includes proof-of-concept experiments on synthetic datasets, as well as experiments on real world datasets, all demonstrating improvement from the proposed technique.
- Nice visual examples demonstrating high and low discrepancy scores (Figure 3a-c).
- Well written and organized.
- Includes many extra details in appendix, including sufficient details for reproducing experiments.
- Work addresses an important limitation of GANs. GANs are notorious for having poor diversity, and recent work has demonstrated many cases of bias in generative models. Being able to improve diversity while still retaining decent image quality is a useful contribution.

Weaknesses:
- Proposed method introduces additional hyperparameters that needs to be tuned (lengths of stage 1 and 2), k value. Not a huge problem since they can be optimized with grid search, but does add an extra level of complexity (and compute/training time) to the solution.
- Missing ablation study of Dia-GAN, specifically Dia-GAN without DRS (i.e., weighted sampling only).
- Missing Precision and Recall metrics for CIFAR-10 and FFHQ 256x256 experiments. Since the goal of this work is to improve diversity, I think it is important to include these metrics for all experiments. Inception Score could be moved to appendix or removed completely as it isn't very informative in this setting.

Questions:
- In Algorithm 1 it appears that the sampling probabilities are set a single time, at the end of stage 1. Have you considered adjusting the sampling probabilities multiple times throughout training, and would this be expected to yield a better final result?
- How would vanilla GAN + DRS perform if combined with the truncation trick [1,2] with truncation values >1? That is, multiply the latent noise by some value >1, like 1.5 or 2 during inference. This method improves sample diversity in exchange for image quality, but I am curious whether combination with DRS would avoid the quality degradation issue. This would be a good simple baseline to compare to.
- In Figure 4, how are the samples so similar? This is unusual for unconditional models. Is the vanilla GAN from stage 1 of Dia-GAN training? If so this should be explained in the caption.
- Are all GANs trained to convergence, or are they stopped early? From what I can tell, the number of iterations for training the GANs in this paper is much less than normal (e.g. ~50k iterations vs 500k iterations in [3] for CIFAR10, 4000k images vs 10000k/25000k images in [4] for FFHQ). This is important because standard GANs generally learn to represent minority samples later in training, so early stopping could produce misleading results.

Other comments/minor concerns:
- In Figure 1e, what does "minority level" mean? Is it the percentage of the dataset composed of minority samples? If so, it would be nice if the graph reflected the actual percentages on the x axis.
- In Figure 1, it would be nice if 1e were on the left side of the image, since single-mode Gaussian is the first one described in the caption (or swap the order in the caption).
- Figure 1, claims that DCGAN was used to train single-mode Gaussian, but appendix says that non-convolutional GAN was used.
- In Figure 3d, the concatenation of RGB channels in the histogram feels weird to me, although I'm not sure if there are any better ways to show this data. Perhaps you could try a separate subplot for each colour channel stacked vertically? Your choice.

Overall, I think this is a decent paper. The proposed technique is reasonable and seems effective, if a bit clunky (requires staged training, additional hyperparameter tuning, and DRS to fix distribution shift). Currently I recommend acceptance, but very weakly. I would be much reassured if the authors would be able to address my questions above, particularly with regards to early stopping.

[1] Brock, Andrew, Jeff Donahue, and Karen Simonyan. "Large scale GAN training for high fidelity natural image synthesis." arXiv preprint arXiv:1809.11096 (2018).
[2] Karras, Tero, Samuli Laine, and Timo Aila. "A style-based generator architecture for generative adversarial networks." arXiv preprint arXiv:1812.04948 (2018).
[3] https://github.com/ajbrock/BigGAN-PyTorch/blob/master/scripts/launch_cifar_ema.sh
[4] https://github.com/NVlabs/stylegan2

**Time Spent Reviewing:**

4.5

---

> ### Author Response · Authors · 2021-08-10
> **Response to Reviewer bY4D**
>
> We sincerely thank the reviewer for the constructive feedback.
> We tried our best to address the reviewer’s concerns and questions. Hope this response answers the reviewer’s questions.
>
> ---------
> **[1. Ablation study of Dia-GAN, specifically Dia-GAN without DRS]**
>
> In Table 2, we compared our method with other baselines followed by DRS and showed that the performance gain is much higher for our score-based sampling method. Nevertheless, we agree with the reviewer that the ablation study of Dia-GAN without DRS is also important for measuring the effect of our sampling method only.
> We evaluated Dia-GAN without DRS for SNGAN on CIFAR-10 and CelebA, and also for StyleGAN2 on FFHQ 256x256.
> For comparison, we also copied the performance measures for vanilla GANs, GOLD [a], and Dia-GAN from Table 2 and 3.
>
> |FID|Vanilla|GOLD [a]|Dia-GAN|Dia-GAN w/o DRS|
> |:----|:----:|:----:|:----:|:----:|
> |CIFAR-10|26.90|28.86|19.66|22.18|
> |CelebA|7.12|7.31|6.70|6.70|
> |FFHQ|14.07|15.53|11.89|11.89|
>
> |Precision|Vanilla|GOLD [a]|Dia-GAN|Dia-GAN w/o DRS|
> |:----|:----:|:----:|:----:|:----:|
> |CIFAR-10|0.58|0.57|0.55|0.55|
> |CelebA|0.68|0.69|0.64|0.65|
> |FFHQ|0.72|0.69|0.69|0.68|
>
> |Recall|Vanilla|GOLD [a]|Dia-GAN|Dia-GAN w/o DRS|
> |:----|:----:|:----:|:----:|:----:|
> |CIFAR-10|0.54|0.54|0.55|0.56|
> |CelebA|0.44|0.44|0.48|0.47|
> |FFHQ|0.27|0.29|0.30|0.31|
>
>
> The results indicate that when we remove DRS from our method, FID gets worse in CIFAR-10, while it remains almost unchanged in larger datasets such as CelebA and FFHQ. For every case, our method without DRS still achieves a better FID compared to the vanilla method or GOLD. The changes in precision and recall from DRS are not significant.
> Therefore, we can conclude that our method itself improves the diversity of the generated samples with a slight drop of precision, and DRS does not give much gain in terms of the overall performance.
>
> ---------
> **[2. Precision and Recall metrics for CIFAR-10 and FFHQ]**
>
> We agree that precision and recall could be better metrics to report than the Inception Score, as our paper is focusing on the diversity in sample generation.
> Please refer to the reported values in the response to Q1.
>
> Similar to the tradeoff relations found in CelebA, we can find the trend on CIFAR-10 and FFHQ as well.
> Our method improves the recall rate but with a slight loss in precision, while it consistently provides the best FID over other baselines.
> Since this trend holds for all the tested GAN variants from simple (fully-connected) to complex GANs (StyleGAN2), and for all different levels of resolution from low 32x32 (CIFAR-10) to high 256x256 (FFHQ), our method is widely applicable to various GAN models and datasets.
>
> We will add the values of precision and recall for CIFAR-10 and FFHQ in the final version of our paper.
>
> ---------
> **[3. Adjusting sampling probabilities multiple times throughout training]**
>
> As the reviewer suggested, we considered adjusting sampling probabilities throughout the training with more than two phases. We conducted the experiments on the Colored MNIST dataset (described in Section 5.3 with performance results in Table 5) with the new score-based sampling method for which the sampling probabilities are adjusted every 1000 steps (thus, the total number of phases is six). We call this new method Adapt-Dia-GAN. We report the Reconstruction Error (RE) score of Adapt-Dia-GAN and compare it with our original Dia-GAN and other baselines (Vanilla and GOLD).
>
> |RE|Vanilla|GOLD [a]|Dia-GAN|Adapt-Dia-GAN|
> |:----|:----:|:----:|:----:|:----:|
> |99%|0.84|0.81|0.22|0.43|
> |95%|0.24|0.30|0.20|0.22|
> |90%|0.22|0.20|0.20|0.21|
>
> This result shows that the adaptive strategy is not as effective as our original method.
> We speculate that this is because the sampling frequency keeps changing before the training is optimized toward the modified sampling frequency, and thus it may result in unstable optimization.
>
> ---------
> **[4. Simple baseline using truncation trick ]**
>
> We thank the reviewer for suggesting a simple and interesting baseline to promote diversity in sample generation. Originally, the truncation trick [e,f] is used to generate high-quality samples by restricting the latent noise to be within the truncation threshold of $\psi <1$. Instead, as the reviewer suggested, we can choose $\psi >1$ to promote diverse sample generation, and then apply DRS to increase fidelity by rejection sampling. We considered this additional baseline by training each model with the usual generator input $z \sim \mathcal{N}(0, I)$ and then generating fake samples with  $z \sim \mathcal{N}(0, {\psi}^{2}I)$ in the inference stage, with $\psi = 1.2$ and $2$. This is equivalent to generating samples $G(\psi z)$ with $z \sim \mathcal{N}(0, I)$.
>
> As shown below, this simple post-hoc method improves the FID score of the original GAN when $\psi=1.2$ for CIFAR-10 dataset. However, the improvement was not as good as our Dia-GAN.
> When $\psi$ gets larger ($\psi = 2.0$), on the other hand, the generated samples start to deteriorate and show a much worse FID score.
>
> |FID|Vanilla|DRS|Trunc($\psi=2.0$)|Trunc($\psi=1.2$)|Dia-GAN (ours)|
> |:----|:----:|:----:|:----:|:----:|:----:|
> |CIFAR-10|26.90|24.54|41.65|24.41|19.66|
> |CelebA|7.12|7.03|51.62|9.31|6.70|
>
> To understand where the performance gain comes from, we also evaluated the precision and recall for this baseline as well.
>
>
> *CIFAR-10*
>
> ||Vanilla|DRS|Trunc($\psi=2.0$)|Trunc($\psi=1.2$)|Dia-GAN (ours)|
> |:----|:----:|:----:|:----:|:----:|:----:|
> |Precision|0.58|0.57|0.45|0.55|0.55|
> |Recall|0.54|0.54|0.50|0.55|0.55|
>
> *CelebA*
>
> ||Vanilla|DRS|Trunc($\psi=2.0$)|Trunc($\psi=1.2$)|Dia-GAN (ours)|
> |:----|:----:|:----:|:----:|:----:|:----:|
> |Precision|0.68|0.68|0.29|0.58|0.64|
> |Recall|0.44|0.44|0.46|0.49|0.48|
>
> We find that the truncation+DRS method improves recall rate, but makes a sharp drop of fidelity, especially for a large $\psi$ value. For example, the precision is only 0.29 for CelebA when $\psi=2$.  On the contrary, our method preserves high fidelity while promoting the diversity in sample generation.
> This clearly supports that emphasizing underrepresented samples during training is crucial to promote diversity in sample generation without degrading the overall quality.
>
> We will add the truncation+DRS method to our baseline in the final version. Thank you for the suggestion.
>
> ---------
> **[5. The generated images in Figure 4]**
>
> We use the majority of the training time for Phase 1 (80% of total steps), so the generator partially converges after Phase 1.
> Therefore, the same latent vector $z$ turns out to give a similar image with details changed after Phase 2 (e.g., wearing sunglasses, or a hat).
>
> ---------
> **[6. Longer training of GANs]**
>
> We agree that our training time was shorter than common practice in GANs.
> Thus, we conducted an evaluation of SNGAN on CIFAR-10 and CelebA with further training of 100k steps, following previous works [b,c,d].
> The reviewer mentioned 500k steps used in BigGAN, but since we used SNGAN which is much smaller than BigGAN, 100k steps were enough to make it converge.
>
> *CIFAR-10*
>
> ||Vanilla|Dia-GAN|Dia-GAN w/o DRS|
> |:----|:----:|:----:|:----:|
> |FID|22.43|	16.49|18.75|
> |IS|7.59|8.10|7.94|
> |Precision|0.57|0.57|0.57|
> |Recall|0.54|0.56|0.56|
>
> *CelebA*
>
> ||Vanilla|Dia-GAN|Dia-GAN w/o DRS|
> |:----|:----:|:----:|:----:|
> |FID|6.83|6.57|6.49|
> |Precision|0.68|0.63|0.65|
> |Recall|0.45|0.49|0.48|
>
> We found similar trends as the previous result (with 50k steps) for longer training of SNGAN on CIFAR-10 and CelebA.
> The overall FID gets better when the model is trained longer, but our method still gives an improvement in terms of FID and recall.
> In addition, we want to point out that our method can offer an efficient way of training, as our method requires much fewer steps to achieve FID better than the best FID of the Vanilla GAN.
>
> ---------
> **[7. The definition of the term `minority level` ]**
>
> The term `minority level` is used to indicate the different percentage of minor samples in the overall dataset, but the composition and configuration of minor subgroups vary between the datasets.
> The exact definition of the `minority level` is available for each dataset in Appendix F.
> For single-mode Gaussian, we use 2D single-mode Gaussian dataset with mean 0 and various covariances $\sigma^2 I$. The minority level 1, 2, 3 in Fig. 1e stands for $\sigma$ = 3, 2.5, 2.
> For Colored MNIST with red (major) and green (minor) samples, and for  MNIST-FMNIST with MNIST (major) and FMINST (minor) samples, we use the majority rate $\rho$ = 90%, 95%, 99% as the minority level 1, 2, 3.
>
> As the reviewer suggested, we will add the explicit definition of the minority level in the main paper and will adjust Figure 1.
>
> ---------
> **[8. GAN used for the single-mode Gaussian experiment]**
>
> We used a simple GAN model with fully-connected layers for the single-mode Gaussian experiment. We apologize for the confusion and thank the reviewer for checking. We will fix it in the final version.
>
> ---------
> [a] Mo et al., Mining gold samples for conditional gans, 2019
>
> [b] Sinha et al., Top-k training of gans: Improving gan performance by throwing away bad samples, 2020
>
> [c] Miyato et al., Spectral normalization for generative adversarial networks, 2018
>
> [d] Tran et al., Self-supervised GAN: analysis and improvement with multi-class minimax game, 2019
>
> [e] Brock et al., Large scale GAN training for high fidelity natural image synthesis, 2018
>
> [f] Karras et al., A style-based generator architecture for generative adversarial networks, 2018

---

### Official Review · Reviewer_n1Zt · 2021-07-16

**Rating:** 6
**Confidence:** 4

**Summary:**

The paper proposes a scoring to measure to what extend an image in a dataset is characterized by an under-represented visual appearance. GANs fail to accurately model these under-represented modes of the data distribution or drop them completely. Here, the score is based on the discriminator predictions on real images, and is used as a basis for score-weighted sampling of real data during training, to show under-represented real images more often. The method is shown to improve mode coverage and to result in better image quality/diversity than the considered baselines.


**Limitations And Societal Impact:**

Limitations and positive societal impact are discussed in section 6, with one sentence each. A negative societal impact is not mentioned, thus it would be good to add the discussion about it to the paper.

**Main Review:**

Strengths

-	The paper studies an important and challenging problem. The ability to better learn underrepresented modes has importance for fairness and is not well studied in the GAN literature.

-	The proposed method falls into the category of sampling and reweighting methods for image generation. It is used to boost underrepresented modes of the data. The main novelty is that this method is applied to real instead of fake samples, the advantage of which is explained on lines 86-89.

-	The theoretical support is provided for the hypothesis that the proposed scoring detects samples from underrepresented modes.

- The method also shows a notable improvement in image quality for a state-of-the-art GAN (StyleGANv2) in Table 3.

Weaknesses:

-	Performing the comparison in Table 2 also for StyleGAN2 and on a more complex dataset with a higher resolution (e.g. ImageNet or FFHQ) would increase the value of the paper. Since state-of-the-art models are of the more interest to the community, and for more complex, higher resolution datasets it is more challenging to capture underrepresented samples. The discriminator would have a much harder job to capture the differences between samples, e.g. semantic classes of ImageNet.

-  The discriminator predictions are used to estimate the discrepancy score, which is used for weighted sampling during training. Thus it would be interesting to see how discriminator network architecture, its design and capacity, influense the performance of the proposed approach. As the score is highly dependent on the discriminator's ability to learn the features which are best describe the data.

-	I’m still doubtful how effective the proposed approach and scores would be on more realistic cases, when some semantic classes are underrepresented. Based on Fig.3 it seems that the high discrepancy score captures more variance in terms of colors and not high-level features. It would be beneficial to see the analysis in Table 1 and Fig.3 on e.g. ImageNet, where number of images per class is quite imbalanced.

-	Most of the experiments are performed using datasets with low-resolution images, exploiting “outdated” GAN models, such as DCGAN, SNGAN, SSGAN.



- The writing and clarity of the paper can be improved:

Line 15 (abstract): It is not very clear what is meant with “minor features” when reading the abstract for the first time.

Line 43/44: What is the difference between minor data instances and under-represented samples? Especially at this point of the text that is not clear.

Section 3.2 and Fig. 1(e): It should be explained in the main text what the “minority levels” stand for, not in the appendix.

The paragraph from line 151-160 is hard to follow and should be described in more detail.

Section 4.4: How does the number of steps in phase 1 affect the outcome? Likewise, is there an optimal number of steps for phase 2?

Section 5.2.: Please specify the resolution of the CIFAR10 and CelebA experiments.

Line 313: “Figure 4 shows examples of generated samples with minor feature appeared by our Dia-GAN.” This sentence should be rephrased.

How was it possible to create almost the same images for SNGAN and Dia-SNGAN in Fig. 4? Even when the latent codes are the same and all seeds are fixed, that is very unlikely to happen. Was the same checkpoint of SNGAN copied after phase 1 and then one copy continued to be trained as SNGAN and the other as Dia-SNGAN in phase 2?

Minor comments:

- Line 26: There is a misplaced dot on this line (“.”)
- Typo in Fig. 3(d): “Historgram”


---
## Post-rebuttal feedback

Thanks to the authors for their feedback and clarifications. Additional experiments and ablations requested by reviewers have been provided, showing the advantages of the proposed approach. I still think it would be beneficial for the paper if the method is also tested on the datasets containing multiple semantic classes, where some of the classes are severely underrepresented, such as ImageNet. Cifar or the faces datasets, such as Celeba or FFHQ, might be too simplistic for this setting. Nevertheless, I think the proposed approach has a potential to be adopted by practitioners and may encourage further research in this direction. Thus I'm more inclined towards acceptance and raise my score to 6.

**Time Spent Reviewing:**

4.5h

---

> ### Author Response · Authors · 2021-08-10
> **Response to Reviewer n1Zt**
>
> We sincerely thank the reviewer for the constructive feedback.
> We tried our best to address the reviewer’s concerns and questions. Hope this response answers the reviewer’s questions.
>
> ---------
> **[1. Performance of our method on StyleGAN2 and higher resolution dataset]**
>
> As shown in Table 3 of our paper, we conducted experiments with StyleGAN2 on the FFHQ (256x256) dataset and demonstrated that our method improves FID by 15% compared to the Vanilla StyleGAN2.
> However, for a more thorough analysis, we assessed other baseline methods in Table 2 with StyleGAN2 as the reviewer suggested, and evaluated each method with more metrics (Precision and Recall).
> In particular, we considered GOLD[a] since it also uses the discriminator output to design a weighting scheme.
> We also compared our method without DRS to check the effectiveness of our own sampling method in boosting the diversity.
>
> ||Vanilla|GOLD [a]|Dia-GAN|Dia-GAN w/o DRS|
> |:----|:----:|:----:|:----:|:----:|
> |FID|14.07|15.53|11.89|11.89|
> |Precision|0.72|0.69|0.69|0.68|
> |Recall|0.27|0.29|0.30|0.31|
>
> We can find similar trends as in Table 2 and observe that our method achieves the best overall quality measured by FID while showing a tradeoff between precision and recall.
> These results imply that our method can be applied to large-scale GANs on high-resolution datasets.
>
> ---------
> **[2. Dependency of our method to the discriminator]**
>
> For the experimental results reported in the main context, we used several different architectures for the discriminator with different capacities. All the detailed architectural configurations are available in Appendix Section F. For example, the number of parameters of discriminators used for each dataset is as follows:
>
> ||25 Gaussian|CIFAR-10|CelebA|FFHQ|
> |:----|:----:|:----:|:----:|:----:|
> |Params|0.13M|1.1M|10.1M|28.8M|
>
> We would like to emphasize that for all the variants of the discriminators, our method consistently provides improved diversity in sample generation.
>
> At the same time, we agree with the reviewer that the discriminator’s capacity may affect the effectiveness of our method in general.
> However, since the discriminator’s capacity eventually governs whether GAN itself can learn the data distribution, as long as GAN achieves reasonable performance in sample generation, our discriminator-based scoring would also be effective.
>
> ---------
> **[3. Effectiveness of our method in capturing semantic (high-level) features]**
>
> First of all, we ask the reviewer to check Figure A2 and A5-6 in the Appendix, which show the images with the lowest / highest discrepancy scores for CelebA dataset.
> We can find that images with the lowest discrepancy scores are mostly white women with blonde hair, which are known to be the majority of the CelebA dataset.
> On the other hand, images with the highest discrepancy scores are mostly men with black hair or wearing hats or glasses which are uncommon in CelebA dataset.
> This shows that our method clearly captures semantic features (e.g., hats, glasses, or gender) that are minor in samples.
> However, some may concern whether these images are indeed underrepresented images that GANs fail to generate with high fidelity.
> To suppress such a concern, in Appendix G.2, we calculated Partial FID (PFID), which evaluates the feature distance between generated samples and only a subset of the training data.
> We calculated PFID with a subset of 5,000 samples having lowest / highest discrepancy scores and denoted them by Low PFID and High PFID, respectively.
> For convenience, we directly copy Table A10 in Appendix G.2 below.
>
> *CIFAR-10*
>
> |Subset|Baseline|Dia-GAN (ours)|
> |:----|:----:|:----:|
> |High PFID|94.64|	77.28|
> |Low PFID|22.43|33.98|
>
> *CelebA*
>
> |Subset|Baseline|Dia-GAN (ours)|
> |:----|:----:|:----:|
> |High PFID|50.25|	42.33|
> |Low PFID|17.25|23.17|
>
> As the results show, the discrepancy score finds samples that GANs are failing to generate with high fidelity.
> Moreover, our method successfully detects and emphasizes such samples, resulting in better PFID for high PFID samples after the training.
>
> In addition, in Table 6 of our main paper, we also conducted a per-attribute analysis of CelebA to thoroughly analyze the ability of our method in capturing semantic features.
> We considered seven different attributes that take only 2-6% portions in the overall CelebA datasets and checked that our discrepancy score is higher by 10-50% for the images with minor attributes compared to the images without them.
> This implies that our score indeed detects and emphasizes semantically meaningful minor attributes.
> As a result, for all minor attributes, both the occurrence rate and partial recall (recall calculated only for samples with each minor attribute) were improved significantly through our method.
>
> To ensure that the ability of our method in capturing semantic features also applies to high-resolution datasets, we considered the FFHQ dataset and classified the race on the FFHQ dataset using the DeepFace architecture [c]. This architecture classifies the images as Asian, Black, Indian, Latino hispanic, Middle eastern, and White. For the FFHQ dataset, Black, Indian, and Middle eastern represent the minority taking less than 5% of the FFHQ dataset.
>
> |Race|Asian|Black|Indian|Latino hispanic|Middle eastern|White|
> |:----|:----:|:----:|:----:|:----:|:----:|:----:|
> |Ratio(%)|19.38|4.80|2.08|10.83|4.04|58.87|
>
> We compared the occurrence rate and partial recall for these minor races after training with vanilla StyleGAN2 and our Dia-GAN, respectively.
>
> |Occurrence(%)|Vanilla|Dia-GAN (ours)|
> |:----|:----:|:----:|
> |Black(4.80%)|3.00|2.99|
> |Indian(2.08%)|0.81|1.16|
> |Middle eastern(4.04%)|3.18|3.49|
>
> |Partial Recall|Vanilla|Dia-GAN (ours)|
> |:----|:----:|:----:|
> |Black|0.27|0.30|
> |Indian|0.26|0.30|
> |Middle eastern|0.27|0.31|
>
> Similar to the results for the CelebA dataset, the occurrence rate and partial recall for minor races in the FFHQ dataset were improved with our method, especially for Indian and Middle-eastern image samples.
>
> In conclusion, this evidence demonstrates that our method successfully captures semantically meaningful minor attributes and emphasizes them during the training, resulting in a diverse generation of minor samples across low- to high-resolution datasets.
>
>
> ---------
> **[4. Experiments with state-of-the-art GANs]**
>
> We believe that the response to the first question (Q1) shows the effectiveness of our method in StyleGAN2, which is one of the state-of-the-art GANs.
> Once again, we would like to emphasize that our method is effective across all the tested GAN variants from simple GANs with fully-connected layers to complex GANs (StyleGAN2), and for all different levels of resolution from low 32x32 (CIFAR-10) to high 256x256 (FFHQ).
>
> ---------
> Points of improvement that the reviewer suggested:
>
> **[5. Clarify the meaning of `minor features` at the beginning of the paper]**
>
> We used the term `minor features` to indicate semantic features that a minor subgroup has, e.g., gender, race, or wearing hats.
> We will clarify the terms we use so that the Abstract itself is self-contained.
>
> ---------
> **[6. Difference between `minor data instances` and `underrepresented samples`]**
>
> Underrepresented samples are samples that GANs fail to generate.
> This includes mode-collapsed samples and samples in minor subgroups that GANs fail to generate with high fidelity.
> The minor data instances are just samples in minor subgroups.
> In our observation, in naive GAN training, the generated samples of minor subgroups tend to have low fidelity and poor diversity.
> The objective of our method is to detect underrepresented samples (which include minor subgroups) and emphasize such samples for better inclusion.
>
> ---------
> **[7. The term `minority level` should be defined in the main paper, not in the Appendix]**
>
> We agree that the definition of the term `minority level` should be available when it is first used.
> We put it in Appendix F.1 and F.3 as the definition comes with the details of each dataset, but we will bring the definition to the main paper in the final version. Thank you for the feedback.
>
> ---------
> **[8. Detailed analysis for Line 151-160 (Analysis of LDRV)]**
>
> We will add more detailed explanations about the relation between LDRV and minor features (also available in Appendix B) in the final version.
>
> ---------
> **[9. The effect of the number of steps in phase 1 and phase 2]**
>
> In Appendix F.8, we showed the change in FID for CIFAR-10 and CelebA datasets with various choices of training steps of Phase 1 and 2.
> For convenience, we copied the results in Table A8 below, showing FID for Dia-GAN with different phase 1 steps (% of total steps).
>
> ||Baseline|20%|40%|60%|80%|
> |:----|:----:|:----:|:----:|:----:|:----:|
> |CIFAR-10|26.90|17.56|16.72|18.65|19.66|
> |CelebA|7.12|6.69|6.90|6.86|6.70|
>
> We can check that our method is not sensitive to the choice of the training steps for Phase 1 and 2. However, there exists an optimal number of steps for Phase 1, which depends on the dataset and GAN model.
>
> ---------
> **[10. The resolution of the CIFAR10 and CelebA]**
>
> For CIFAR-10, we use the image size of 32x32, and for CelebA, 64x64. Please note that this is the same setting with other baseline papers [a,b].
>
> ---------
> **[11. The generated images in Figure 4]**
>
> We use the majority of the training time for Phase 1 (80% of total steps), so the generator partially converges after Phase 1.
> Therefore, the same latent vector $z$ turns out to give a similar image with details changed after Phase 2 (e.g., wearing sunglasses, or a hat).
>
> ---------
>
> [a] Mo et al., Mining gold samples for conditional gans, 2019
>
> [b] Sinha et al., Top-k training of gans: Improving gan performance by throwing away bad samples, 2020
>
> [c] Taigman et al., Deepface: Closing the gap to human-level performance in face verification, 2014

---

### Official Review · Reviewer_FXfS · 2021-07-20

**Rating:** 7
**Confidence:** 4

**Summary:**

This submission addresses the problem of limited diversity in GANs due to not modeling underrepresented samples. The authors propose two metrics based on the log-density ratio (LDR), namely, its mean and variance over time. This allows the model to monitor underrepresented and underrepresented samples. Then, the authors propose a training method where data is sampled according to an heuristic based on those two metrics. The results outperform other related methods used for the same purpose, across different architectures and datasets.

**Limitations And Societal Impact:**

The authors mention the in the conclusion. I think the authors should also include some words on the misuse of GANs.

**Main Review:**

Originality:
The method is built on top of previous GAN frameworks and the LDR statistic, which are described clearly in the submission. The two metrics proposed are straightforward yet effective. Score-based sampling is also not novel, although the authors propose also a simple yet effective method.

Quality:
The submission is sound, and the claims are supported by experiments with different GAN frameworks in both synthetic and realistic datasets.

Clarity:
The organization and writing are clear. The toy examples with synthetic datasets are very illustrative.

Significance:
The idea is sound, yet borrowing heavily from previous works, and achieves a moderate gain in most experiments. Other practicioners can benefit from this approach. Similarly, the paper also includes useful insights.

**Needs Ethics Review:**

Yes

**Time Spent Reviewing:**

5

---

> ### Author Response · Authors · 2021-08-10
> **Response to Reviewer FXfS**
>
> We sincerely thank the reviewer for the constructive feedback.
> We would like to comment on the possible societal impact of misuse of GANs.
>
> ---------
> **[1. The societal impact of misuse of GANs]**
>
> High-fidelity GANs can be used to generate fake data. Recently, generating DeepFake videos using GANs has become problematic and it has been even considered a serious cybercrime [a].
>
> Moreover, the issue of biased sample generation of GANs can also cause serious social issues of discrimination.
> Our work may partly solve the problem of biased sample generation and give a solution for the better inclusion of minorities.
>
>
> ---------
> [a] Nguyen et al., Deep Learning for Deepfakes Creation and Detection, 2019

---

### Review · Ethics_Reviewer_FGY7 · 2021-07-30

**Recommendation:** The authors should address the misuse…

**Ethical Issues:**

Yes

**Ethics Review:**

The reviewer who flagged for ethical review indicated that the authors should discuss the misuse of GANs.

---

> ### Author Response · Authors · 2021-08-16
> **Response to Ethics Reviewer FGY7**
>
> As we responded to the Reviewer FXfs, we would like to comment on the possible societal impact of misuse of GANs.
>
> ---------
> **[1. The societal impact of misuse of GANs]**
>
> High-fidelity GANs can be used to generate fake data. Recently, generating DeepFake videos using GANs has become problematic and it has been even considered a serious cybercrime [a].
>
> Moreover, the issue of biased sample generation of GANs can also cause serious social issues of discrimination.
> Our work may partly solve the problem of biased sample generation and give a solution for the better inclusion of minorities.
>
> We will add the discussion about the misuse of GANs in the final version.
>
>
> ---------
> [a] Nguyen et al., Deep Learning for Deepfakes Creation and Detection, 2019

---

### Review · Ethics_Reviewer_3ftZ · 2021-08-12

**Recommendation:**

Yes, definitely possible to address in the current version, by adding to the discussion section.

**Ethical Issues:**

Yes

**Ethics Review:**

The authors describe an approach for identifying and emphasizing images in the dataset that contain features that are underrepresented in that dataset. Examples from the CelebA dataset include individuals wearing hats or members of racial minorities.


Their approach identifies statistically-divergent inputs and weights these in the model to increase the ability of the model to identify inputs with these features. This has the effect of tagging and emphasizing input images with, for example, people wearing hats (and presumably other head coverings) and members of racial minorities.

The discussion has a passing mention that including these 'minor samples' may assist with incorporating minorities into machine learning. I would suggest expanding this with a few sentences about potential uses and misuses. This would allow them to address issues around the ethical use and potential downstream effects of their approach such as: Does this approach change how we would need to assess models for bias? Could it mask biases in source datasets? It appears from the paper that this approach improves on training on non-inclusive datasets, but how does it compare to training on datasets that adequately cover features of interest? If hats, for example, are important isn't it better to include an adequate number of pictures including hats in the training dataset? Could dataset curators use this approach to avoid collecting images or other data that is actually representative and instead argue that these techniques allow skewed datasets to be used to train models intended for use on more general populations?

Addressing these issues should not require substantial changes to the paper; I would suggest expanding the Discussion section with perhaps a new paragraph.

---

> ### Author Response · Authors · 2021-08-16
> **Response to Ethics Reviewer 3ftZ**
>
> We sincerely thank the reviewer for providing valuable aspects regarding the possible impacts of our method. We will add a paragraph describing potential uses and misuses of our method in the discussion section.
>
> When the training dataset is skewed, vanilla GAN models tend to underrepresent the minor subgroup (i.e., minor subgroup samples appear in much less occurrence than the original percentage) and generate minor subgroup samples with low fidelity.
> Our method solves this problem by proposing a discrepancy score that can distinguish such an underrepresented minor subgroup.
>
> As the reviewer pointed out, the proposed discrepancy score can be used in multiple positive ways, but at the same time, it can be misused for wrong purposes.
> On the good side, the ability of our score in detecting minor samples from unlabeled datasets could be further expanded and used for utilizing skewed datasets to train models representing more balanced datasets, by adding a hyperparameter that can tune the level of emphasis for underrepresented samples.
> On the other hand, an abuser might instead be able to remove such minor subgroup samples and deteriorate the bias in sample generation.
>
> More detailed answers for the reviewer’s points are addressed below.
>
> ---------
> **[1. Does this approach change how we would need to assess models for bias? Could it mask biases in source datasets?]**
>
> Our method tries to train a generative model producing a model distribution close to the data distribution with the same composition of attributes as the data distribution has. Even though our method can detect and emphasize underrepresented samples, we don’t believe that our proposed approach can fully remove the biases in the sample generation.
> As shown in Table 6 of our main paper, even though the occurrence rate increases for minor attributes than the Vanilla GANs, it still does not reach the original percentage of the minor attribute. However, we would like to emphasize that our approach can be further extended to control how much the minor samples are emphasized and may possibly reduce biases in sample generation, by adding a hyperparameter that can tune the level of emphasis for underrepresented samples in the score-based weighted sampling.
>
> ---------
> **[2. How does the DiaGAN improve training on the non-inclusive dataset compared to training on datasets that adequately cover features of interest? ]**
>
> We cannot say that our approach gives better performance than GANs trained on the meticulously curated dataset. However, it costs a lot to carefully collect or manipulate datasets with proper portions for each important feature. Moreover, certain subgroup samples can be difficult to collect. If the major subgroup samples are removed to match the number of samples in each subgroup, the quality of the major subgroup can be degraded. On the contrary, our method provides a simple yet effective way to diagnose underrepresented samples even from unlabelled datasets during training of GANs and to properly emphasize those samples, which increases the diversity in sample generation.
>
> ---------
> **[3. Could dataset curators use this approach to avoid collecting images or other data that is actually representative and instead argue that these techniques allow skewed datasets to be used to train models intended for use on more general populations? ]**
>
> Our approach might not completely remove the necessity of collecting samples with demographic parity. However, our method can be a proper solution when the user is trying to use an already existing unbalanced dataset to train models with better coverage for minor attributes.

---

### Decision · Program_Chairs · 2021-09-27

**Decision:**

Accept (Poster)

**Comment:**

The paper addresses the problem of mode-dropping of underrepresented areas of the data manifold in the context of GANs. The main result relies on a reweighing scheme for the real data, which is novel according to the reviewers.

Two reviewers think the paper is clear and well written while the third one believes the clarity could be improved.

One recurring comment is that the experimental section could be stronger by using more recent models and larger datasets.

In general, the reviewers tend to agree that this is an incremental progress, based on other papers, but all agree that it is important enough to be published.

Ethics: The reviewers only highlight general possible ethics issues which are common to most GAN (and generative models). The authors did acknowledge some of these issues in the paper and their answers to the reviewers, along to the planned editing, are satisfactory.

Given all that, and the general consensus for acceptance, I recommend this paper to be published as a poster.